# GDEGAN: GAUSSIAN DYNAMIC EQUIVARIANT GRAPH ATTENTION NETWORK FOR LIGAND BINDING SITE PREDICTION

## ABSTRACT

Accurate prediction of binding sites of a given protein, to which ligands can bind, is a critical step in structure-based computational drug discovery. Recently, Equivariant Graph Neural Networks (GNNs) have emerged as a powerful paradigm for binding site identification methods due to the large-scale availability of 3D structures of proteins via protein databases and AlphaFold predictions. The state-of-the-art equivariant GNN methods implement dot product attention, disregarding the variation in the chemical and geometric properties of the neighboring residues. To capture the variation in properties, we propose GDEGAN (Gaussian Dynamic Equivariant Graph Attention Network), which replaces simple dot-product attention with adaptive kernels that recognize binding sites. The proposed attention mechanism captures variation in neighboring residues using statistics of their characteristic local feature distributions. Our mechanism dynamically computes neighborhood statistics at each layer, using local variance as an adaptive bandwidth parameter with learnable per-head temperatures, enabling each protein region to determine its own context-specific importance. Our model shows better predictive performance, outperforming existing methods with relative improvements of 37-66 % in DCC and 7-19 % DCA success rates across COACH420, HOLO4k, and PDBBind2020 datasets. These advances have direct application in accelerating protein-ligand docking by identifying potential binding sites for therapeutic target identification.

## 1 INTRODUCTION

The functional behavior of proteins is governed mainly by their interaction with other molecules to modulate their function, such as small-molecule ligands (Du et al., 2016). These interactions are precise, occurring at well-defined geometric and chemical regions on the protein surface known as binding sites or pockets. Precisely predicting potential binding sites based on a protein's 3D structure is a foundational challenge in rational drug design (Zheng et al., 2013) and structural biology (Schomburg et al., 2014). The success of models like AlphaFold (Jumper et al., 2021; Abramson et al., 2024) in accurately predicting protein 3D structures has significantly advanced the capabilities of structure-based drug design methodologies (Tunyasuvunakool et al., 2021; Sadybekov & Katritch, 2023). Within this field, it is crucial to differentiate between two complementary computational tasks. The first, *protein-ligand binding sites* identification from 3D structures of proteins, is the fundamental challenge of discovering surface pockets capable of binding novel or unknown ligands. This is particularly vital for the majority of proteins with no known ligands or binding partners. The second, *docking* (Zhang et al., 2023; Stärk et al., 2022; Lu et al., 2022) builds upon this by predicting the precise binding pose and orientation of a known ligand within a target site. While historically these have been formidable challenges, both have recently seen significant progress driven by the application of geometric deep learning (Stärk et al., 2022; Cai et al., 2024; Lu et al., 2022; Zhang et al., 2023; Hussein et al., 2015; Gainza et al., 2020; Méndez-Lucio et al., 2021; Ganea et al., 2022; Satorras et al., 2021; Zhang et al., 2024; Schütt et al., 2018), leading to powerful new tools for drug discovery. Additionally, a complementary line of research focuses on binding affinity prediction for known protein-ligand pairs (Somnath et al., 2021; Cai et al., 2024), which ranks the strength of interactions rather than identifying binding locations.

**Ligand Binding Site (LBS) Identification.** Over the years, various computational methods from classical machine learning to deep learning have been successfully employed for LBS identification, combining proteins' physical, chemical and geometric information. Earlier approaches, including P2Rank (Krivák & Hoksza, 2018), a random-forest based technique that utilized protein surface information, and Fpocket (Le Guilloux et al., 2009), which depended on Voronoi tessellation and alpha spheres (Liang et al., 1998b) for efficacy, are constrained by the limited expressivity of protein representation. Early applications of deep learning to this problem, pioneered by Convolution Neural Networks (CNNs) (LeCun et al., 2002), led to various methods including DeepSite (Jiménez et al., 2017), DeepPocket (Aggarwal et al., 2021) and DeepSurf (Mylonas et al., 2021) treating proteins as 3D volumetric data and applying 3D CNNs to predict binding regions. However, these voxel-based approaches under-perform due to their fundamental misalignment with the irregular, sparse nature of protein structures. More importantly, they are sensitive to the protein's orientation in 3D space (Zhang et al., 2024). These limitations motivated representing proteins more naturally as graphs, where the atoms or residues serve as nodes and the interactions between them are the edges. This perspective is perfectly suited for Graph Neural Networks (GNNs), and in particular, equivariant GNNs (Satorras et al., 2021), which have become the standard for 3D geometric deep learning. By design, these models respect the rotational and translational symmetries of the physical world, directly addressing the key failure of CNNs. Consequently, modern methods like EquiPocket (Zhang et al., 2024), which utilize EGNN (Satorras et al., 2021) as backbone, have proven to be powerful for LBS identification.

**Equivariant Graph Neural Networks.** Equivariant graph neural networks have progressed in two directions: scalarization-based models (Satorras et al., 2021; Schütt et al., 2018; 2021; Du et al., 2023) and high-degree steerable models (Batzner et al., 2022; Batatia et al., 2022; Musaelian et al., 2023; Qiao et al., 2022; Liao & Smidt, 2023; Liao et al., 2024). The scalarization-based models function by transforming 3D data, such as coordinates, into scalar characteristics (e.g., distance), hence enhancing computational efficiency and scalability. Nonetheless, their expressivity is constrained in contexts such as protein data modeling, where the comprehension of symmetry and spatial relationships is essential for capturing geometric patterns. In contrast, high-degree steerable models operate directly on rich geometric features (irreducible representations) via the Clebsch-Gordan product, preserving essential spatial relationships. Despite their robust theoretical foundation and superior performance, they are computationally intensive, particularly for larger graphs such as proteins Cen et al. (2024). GotenNet (Aykent & Xia, 2025) presents a solution that balances expressiveness and computing efficiency by implementing a spherical-scalarization model. Building on this, we adopt GotenNet (Aykent & Xia, 2025) as the backbone for our model, applying its efficient framework for processing higher-degree features to the protein-ligand binding site identification task.

**Yet even Equivariant GNNs applied to proteins exhibit a critical limitation.** While recent E(3)-equivariant methods, such as EquiPocket, the current state-of-the-art (Zhang et al., 2024) method, have significantly advanced ligand binding site identification. Their message-passing frameworks are inherently based on a static, context-agnostic attention mechanism known as dot-product attention or self-attention, formally expressed as $\alpha_{ij} \propto exp(f(h_i, h_j))$, where $f$ quantifies similarity. These methods employ a globally fixed similarity metric to assess inter-atomic importance, which is fundamentally misaligned with the nature of proteins, as they are characterized by extreme structural and chemical heterogeneity.

**Our Approach.** Our approach is motivated by the key insight that binding sites often appear as statistically distinct, tightly clustered regions compared to the rest of the protein surface. Exploiting this property, we overcome the aforementioned limitations by replacing dot-product attention with dynamic, context-aware statistical fitting. Inspired by recent work in probabilistic attention for non-geometric modalities (Ioannides et al., 2024), we introduce Gaussian Dynamic Attention mechanism, adapted for the first time to equivariant graph representations. We build upon GotenNet architecture (Aykent & Xia, 2025) as it already handles the high-degree steerable features efficiently. Our proposed model GDEGAN, computes attention scores by measuring how statistically probable a neighboring atom's features are, given a Gaussian distribution defined by the target atom's local neighborhood. By dynamically computing the mean and variance of each atom's local environment at every layer, our attention mechanism becomes inherently adaptive. This adaptation to the emergent local geometry of the protein graphs provides a more powerful and physically grounded inductive bias, enabling the model to learn more robust representations of complex protein structures.

**Contributions.** In this work, we aim to improve on finding the most probable binding candidates for LBS identification task by assigning dynamic and context-aware attention. To this end, we make the following contributions:

- We introduce Gaussian Dynamic Attention mechanism that characterizes each atom's neighborhood using learnable Gaussian parameters. This design preserves E(3)-equivariance by computing attention from invariant local statistics.

- We investigate the use of high-degree steerable E(3)-Equivariant GNNs to the critical task of protein-ligand binding site identification, demonstrating their effectiveness in capturing complex geometries.

- We demonstrate through extensive experiments that our proposed GDEGAN surpasses state-of-the-art methods on multiple benchmarks and achieves a significant improvement in inference speed, validating the efficacy and efficiency of our adaptive attention design.

## 2 PRELIMINARIES

**Protein Graph Representation.** We represent a protein structure as a geometric graph $\mathcal{G} = (\mathcal{V}, \mathcal{E}, \mathcal{P})$, where $\mathcal{V}$ denotes the set of $N$ residues, $\mathcal{E}$ denotes edges between spatially proximate residues, and $\mathcal{P} = \{\mathbf{p}_i \in \mathbb{R}^3\}_{i=1}^N$ represents the 3D coordinates of $C_\alpha$ atoms. Each node $v_i \in \mathcal{V}$ is characterized by it's scaler features $h_i \in \mathbb{R}^{n_d}$ and equivariant features $\tilde{\mathbf{X}}^{(l)} \in \mathbb{R}^{(2l+1) \times h_d}$ of degree $l \in \{1, ..., L_{\max}\}$. Edges connect residues within a spatial cutoff: $\mathcal{E} = \{(i, j) : \|\mathbf{p}_i - \mathbf{p}_j\| < r_c, i \neq j\}$, where $r_c$ is set to 10 Å for capturing relevant interactions and $n_d$ is initial node feature dimension and $h_d$ denotes node embedding hidden dimensions.

To initialize nodes with dimension $n_d$, we use pre-trained ESM-2 embeddings $h_i \in \mathbb{R}^{n_d}$, which are then projected to hidden-dimensional features $h_d$ using learned transformations. These embeddings capture sequence context and evolutionary patterns needed to identify binding sites. Nodes are labeled binding ($y_i = 1$) or non-binding ($y_i = 0$) based on closeness to ligand atoms during training detailed on Appendices A.1 and A.2.

**Equivariance and Invariance.** In 3D geometric learning, the symmetry of physical laws necessitates that models adhere to the Euclidean group E(3), which includes rotations, reflections, and translations. A function $f$ is E(3)-equivariant if for rotation/reflection $\mathbf{R} \in O(3)$ and translation $\mathbf{t} \in \mathbb{R}^3$, it satisfies $f(g \cdot \mathbf{P}, h) = g \cdot f(\mathbf{P}, h)$, where $g \cdot \mathbf{P} = \mathbf{RP} + \mathbf{t}$ denotes the group action on positions $\mathbf{P}$, and invariant features $h$. Invariant functions satisfy: $f(g \cdot \mathbf{P}, h) = f(\mathbf{P}, h)$, producing unchanged outputs for scalar quantities.

**Task Formulation.** Given a geometric protein graph $\mathcal{G} = (\mathcal{V}, \mathcal{E}, \mathcal{P})$, we formulate the task as learning an equivariant model $f(\mathcal{G}, y)$ to predict the binding probabilities $\hat{y}_i \in [0, 1]$ for each residue $v_i \in \mathcal{V}$.

## 3 GDEGAN: GAUSSIAN DYNAMIC EQUIVARIANT GRAPH ATTENTION NETWORK

We enhance GotenNet (Aykent & Xia, 2025) by incorporating a Gaussian dynamic attention module. Although GotenNet attains superior performance in molecular property prediction via equivariant tensor attention, its uniform attention approach to atomic neighborhoods constrains its efficacy in predicting protein-ligand binding sites, where geometric heterogeneity plays a crucial role. To address this limitation, we leverage the observation of binding sites having different geometric patterns, with varying curvature and chemical properties. This motivates three key modifications: statistical attention that adapts to local variance, protein-specific embeddings, and directed supervision. The complete architecture of GDEGAN is illustrated in Figure 1.

### 3.1 EQUIVARIANT GEOMETRIC TENSORS

Here, we describe different representations for both scalars and tensors. Tensor representations are initialized using spherical harmonics to capture spatial information from rank 0 to $L_{max}$. Edge

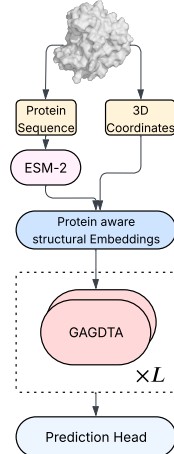 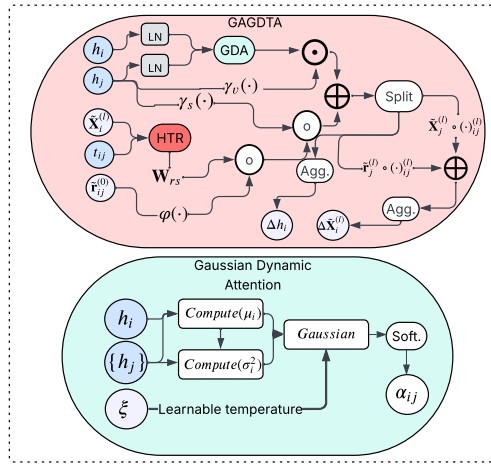

Figure 1: **GDEGAN architecture for protein ligand binding site identification. Left:** Overview of the GDEGAN framework showing the integration of protein-specific ESM-2 embeddings with geometric processing through $L$ layers of Gaussian Dynamic Attention. **Right:** Detailed view of the Gaussian Dynamic Attention (GDA) module. In this $\oplus$, $\cdot$ and $\circ$ denotes addition, dot product and element-wise product respectively. HTR is inherited from GotenNet (Aykent & Xia, 2025). Soft. stands for Softmax, Agg. for Aggregation.

geometry is encoded via spherical harmonics: $\tilde{\mathbf{r}}_{ij}^{(l)} = Y^{(l)}(\hat{\mathbf{r}}_{ij})$ where $\hat{\mathbf{r}}_{ij} = (\mathbf{p}_i - \mathbf{p}_j)/\|\mathbf{p}_i - \mathbf{p}_j\|_2$ is the unit vector and $Y^{(l)} : S^2 \rightarrow \mathbb{R}^{2l+1}$ denotes the degree-$l$ spherical harmonics that map the unit sphere to a $(2l + 1)$ dimensional vector. These basis functions enable the network to process geometric information while preserving equivariance. For $l = 0$, $\tilde{\mathbf{r}}_{ij}^{(0)}$ : scalar invariants; $l = 1$, $\tilde{\mathbf{r}}_{ij}^{(1)}$: directional vectors; $l = 2$, $\tilde{\mathbf{r}}_{ij}^{(2)}$: quadrupole moments. Where, $\tilde{\mathbf{r}}_{ij}^{l}$ is initialized based on their relative positions $\mathbf{p}_i$ and $\mathbf{p}_j$ of nodes $i$ and $j$ in increasing order of geometric complexity, specifically $\tilde{\mathbf{r}}_{ij} = \{\tilde{\mathbf{r}}^{(0)}, \tilde{\mathbf{r}}^{(1)}, ..., \tilde{\mathbf{r}}^{(L_{max})}\}$ and $\tilde{\phantom{r}}$ represents the steerable features. Node features comprise of invariant scalars $h \in \mathbb{R}^{n_d}$ and equivariant high degree steerable features $\tilde{\mathbf{X}}^{(l)}$. These features transform predictably under E(3) operations, with scalars remaining invariant and steerable features transforming according to their degree $l$.

## 3.2 GEOMETRY-AWARE PROTEIN EMBEDDINGS

Unlike standard molecular GNNs that rely solely on atomic properties, our approach leverages pretrained protein representations while incorporating spatial relationships through geometric information, allowing efficient message passing for both nodes and edges. Given pre-computed ESM-2 (Lin et al., 2022) embeddings for each residue $h_i \in \mathbb{R}^{n_d}$, we construct initial node features through a two-stage process that combines evolutionary information with local structural context. We use ESM-2 (Lin et al., 2022) embeddings as they provide with sequential information orthogonal to our geometric processing, unlike structure aware alternatives like ProstT5 (Heinzinger et al., 2024) that would create geometric features redundancy (Mallet et al., 2025).

**Neighborhood-Aware Message Aggregation.** For each residue $i$, we aggregate information from spatial neighbors:

$$\mathbf{m}_i = \sum_{j \in \mathcal{N}(i)} \mathbf{W}_a(h_j) \circ \left( \phi(\tilde{\mathbf{r}}_{ij}^{(0)}) \mathbf{W}_{\text{rbf}} \right) \circ \varphi(\tilde{\mathbf{r}}_{ij}^{(0)}) \tag{1}$$

where $\mathcal{N}(i) = \{j : \|\mathbf{p}_i - \mathbf{p}_j\| < r_c, j \neq i\}$ defines the spatial neighborhood, $\phi : \mathbb{R} \rightarrow \mathbb{R}^K$ represents $K$ radial basis functions encoding distances, $\mathbf{W}_{\text{rbf}} \in \mathbb{R}^{K \times h_d}$ projects RBF features, $\mathbf{W}_a \in \mathbb{R}^{n_d \times h_d}$ projects node evolutionary features, $\circ$ denotes element-wise product and $\varphi(\tilde{\mathbf{r}}_{ij}^{(0)})$ is a smooth cutoff function ensuring differentiability.

**Context-Enriched Feature Construction.** The final node node features combining self and neighborhood information:

$$h_i^{(0)} = \mathbf{W}_u \left( \sigma \left( \text{LN} \left[ \mathbf{W}_h(h_i \| \mathbf{m}_i) \right] \mathbf{W}_d \right) \right) \tag{2}$$

where $\|$ denotes concatenation, LN is layer normalization for training stability, $\sigma$ is the activation function, and $\mathbf{W}_h, \mathbf{W}_d, \mathbf{W}_u$ are learned projections. This formulation enables each residue to incorporate both its evolutionary signature and local structural environment.

**Geometry-Aware Edge Scaler Features.** Edge scaler features are initialized to capture pairwise relationships enhanced by spatial information:

$$t_{ij}^{(0)} = (h_i^{(0)} + h_j^{(0)}) \circ \left( \phi(\tilde{\mathbf{r}}_{ij}^{(0)}) \mathbf{W}_e \right) \tag{3}$$

This symmetric formulation guarantees that $\mathbf{t}_{ij} = \mathbf{t}_{ji}$, preserving consistency in undirected protein graphs while integrating distance-dependent modulation via RBF-encoded spatial characteristics. Here, $\mathbf{t}_{ij}^{(0)} \in \mathbb{R}^{e_d}$ represents the edge features of dimension $e_d$, and $\mathbf{W}_e$ denotes a learned transformation matrix.

**High Degree Equivariant Steerable features Initialization.** These high degree steerable features that capture complex geometric information are initialized as $\mathbf{0}$ initially and are updated during attention aware feature update module, which will be discussed in the later section.

$$\tilde{\mathbf{X}}_i^{(l),(0)} = \mathbf{0} \in \mathbb{R}^{(2l+1) \times h_d}, \quad \forall l \in \{1, ..., L_{\max}\} \tag{4}$$

### 3.3 GEOMETRY AWARE GAUSSIAN DYNAMIC TENSOR ATTENTION

Unlike standard dot-product attention that treats all nodes uniformly, protein binding sites exhibit distinct geometric and chemical patterns: binding pockets are characterized by clustered residues with specific spatial arrangements, while other surface regions show more dispersed distributions (Krivák & Hoksza, 2018). We hypothesize that high local chemical diversity translates to a high variance in a learned feature space of the surrounding residue. Therefore, local feature variance acts as a reliable signal for identifying functionally significant transition areas. This hypothesis is grounded in prior observations that binding pockets exhibit geometric and chemical heterogeneity (Liang et al., 1998a), a property implicitly exploited by successful classical methods like P2Rank (Krivák & Hoksza, 2018) through chemical diversity descriptors (detailed analysis in Appendix E). Our Gaussian Dynamic Attention exploits this inherent heterogeneity by computing local neighborhood statistics $(\mu_i, \sigma_i^2)$ from the ESM-2 features $h_i$, enabling adaptive attention that responds to the local geometric context. The variance $\sigma_i^2$ modulates attention weights: high variance amplifies attention to capture complex binding site boundaries, while lower variance reduces unnecessary focus. Further, it is enhanced by incorporating spatial information. Specifically, for each residue $i$, with neighborhood $\mathcal{N}(i)$ we compute:

$$\mu_i = \frac{1}{|\mathcal{N}(i)|} \sum_{j \in \mathcal{N}(i)} h_j \tag{5}$$

$$(\sigma_i)^2 = \frac{1}{|\mathcal{N}(i)|} \sum_{j \in \mathcal{N}(i)} (h_j - \mu_i)^2 \tag{6}$$

$$d_{ij}^{\text{scaled}} = \frac{h_j - h_i}{\sqrt{\sigma_i^2 + \epsilon}} \tag{7}$$

These statistics provide a distributional summary of the local neighborhood, capturing both the central tendency and spread of features. Where $d_{ij}$ is dimension wise normalization, with stability factor $\epsilon$ (Appendix D for detailed training stability analysis).

**Learnable Gaussian Parameters.** We introduce $H$ learnable variance parameter $\xi$ that controls the temperature of attention for each of the $H$ attention heads. This parameter adaptively modulates the sensitivity of attention to feature differences, allowing each head to specialize in different scales of molecular interactions. For molecular graphs, we compute the Gaussian attention score directly from pairwise node differences:

$$\alpha_{ij} = \frac{\exp\left(-\frac{(\|d_{ij}^{\text{scaled}}\|_2)^2}{2\xi}\right)}{\sum_{k \in \mathcal{N}(i)} \exp\left(-\frac{(\|d_{ik}^{\text{scaled}}\|_2)^2}{2\xi}\right)} \tag{8}$$

where $\|d_{ij}^{\text{scaled}}\|_2$ is the L2 norm of the variance-normalized feature differences.

**Integrating with Equivariant Features.** To combine gaussian dynamic attention with equivariant features we follow GotenNet (Aykent & Xia, 2025) framework for updating both scalar and steerable features. Given attention scores $\alpha_{ij}$ from equation 8 we compute attention-weighted messages and combine them with geometric encoding and then splits into $S$ components:

$$\mathbf{o}_{ij} = \alpha_{ij} \cdot \gamma_v(h_j) + (t_{ij}\mathbf{W}_{rs}) \circ \gamma_s(h_j) \circ \varphi(\tilde{\mathbf{r}}_{ij}^{(0)}) \tag{9}$$

$$\{o_{ij}^s, \{o_{ij}^{d,(l)}\}_{l=1}^{L_{\max}}, \{o_{ij}^{t,(l)}\}_{l=1}^{L_{\max}}\} = \text{split}(\mathbf{o}_{ij}, h_d) \tag{10}$$

Here $\gamma_v, \gamma_s : \mathbb{R}^{h_d} \to \mathbb{R}^{S \cdot h_d}$ are MLPs and $\mathbf{W}_{rs} \in \mathbb{R}^{e_d \times (S \cdot h_d)}$ is a learnable weight matrix. $S$ is multiplying factor to generate different coefficients for different $l$ values calculated as $1 + 2 \times L_{max}$. Finally, features are updated maintaining equivariance:

$$\Delta h_i = \oplus_{j \in \mathcal{N}(i)}(o_{ij}^s) \tag{11}$$

$$\Delta \tilde{\mathbf{X}}_i^{(l)} = \oplus_{j \in \mathcal{N}(i)} \left[ o_{ij}^{d,(l)} \circ \tilde{\mathbf{r}}_{ij}^{(l)} + o_{ij}^{t,(l)} \circ \tilde{\mathbf{X}}_j^{(l)} \right] \tag{12}$$

Here, each degree $l \in [1, L_{max}]$ contributes its own component and representations of residues are updated as follows:

$$h_i \leftarrow h_i + \Delta h_i, \quad \tilde{\mathbf{X}}_i^{(l)} \leftarrow \tilde{\mathbf{X}}_i^{(l)} + \Delta \tilde{\mathbf{X}}_i^{(l)} \tag{13}$$

By replacing uniform dot-product attention with geometry-aware Gaussian attention that adapt to local feature distributions, GAGDTA enables precise discrimination between binding pockets and surface regions. This adaptive mechanism proves particularly effective for ligand binding site prediction, where the inherent heterogeneity of protein surfaces from tightly clustered binding pockets to dispersed surface residues demands context-aware attention patterns, as demonstrated in our experiments Section 4.

### 3.4 Hierarchical Processing and Equivariant Refinement

Following the GAGDTA layer, we adopt GotenNet (Aykent & Xia, 2025) hierarchical tensor refinement (HTR) and equivariant feed-forward (EQFF) modules with minimal modifications to maintain architectural consistency. To summarize:

**Edge refinement via HTR.** Edge features are refined using inner products of high-degree steerable features:

$$\mathbf{w}_{ij} = \text{Agg}_{l=1}^{L_{\max}} \langle \tilde{\mathbf{X}}_i^{(l)}\mathbf{W}_q, \tilde{\mathbf{X}}_j^{(l)}\mathbf{W}_k^{(l)} \rangle, \quad t_{ij} \leftarrow t_{ij} + \gamma_w(\mathbf{w}_{ij}) \circ \gamma_t(t_{ij}) \tag{14}$$

where $\mathbf{w}_{ij}$ is aggregated similarity between node $i$ and $j$, $\mathbf{W}_q, \mathbf{W}_k \in \mathbb{R}^{e_d \times e_d}$ are tensor projection matrices where, $\mathbf{W}_q$ is shared across degree $l \in [1, ..L_{max}]$ and $\mathbf{W}_k^{(l)}$ is degree specific. $\gamma_w : \mathbb{R}^{e_d} \to \mathbb{R}^{e_d}$ and $\gamma_t : \mathbb{R}^{e_d} \to \mathbb{R}^{e_d}$ are MLPs.

This refinement enriches edge representations with geometric information extracted from steerable features, enhancing the model's ability to capture spatial relationships between binding and non-binding residues.

**Channel Mixing via EQFF.** This is employed after GADGTA for efficient channel wise interaction while preserving equivariance:

$$\text{EQFF}(h, \tilde{\mathbf{X}}^{(l)}) = \left( (h + m_1) || (\tilde{\mathbf{X}}^{(l)} + m_2 \circ \tilde{\mathbf{X}}^{(l)}\mathbf{W}_v) \right) \tag{15}$$

where $(m_1, m_2) = \text{split}(\gamma_m((\|\tilde{\mathbf{X}}^{(l)}\mathbf{W}_v\|_2)||h))$ are modulation factors computed from feature norms. $\mathbf{W}_v$ is learnable weight matrices, $\| \cdot \|_2$ denotes $L_2$ norm and $\gamma_m$ is and MLP.

### 3.5 Theoretical Properties of GDEGAN

GotenNet (Aykent & Xia, 2025) proves in Appendices A and B that the end-to-end architecture is $E(3)$-equivariant. By introducing ESM-embeddings (Lin et al., 2022) as node features, we establish the following:

**Proposition 3.1** (From $E(3)$ to $SE(3)$ Equivariance with Invariant Node Features). *The $E(3)$ equivariance breaks after the introduction of ESM-embeddings because, now node features do not encode chirality information. Therefore, the network maintains $SE(3)$ equivariance but loses reflection equivariance.*

*Proof in Appendix B.1.*

**Proposition 3.2** (GDA Preserves $SE(3)$ Equivariance). *The Gaussian Dynamic Attention mechanism, when applied to invariant scalar features from ESM-embeddings, preserves the $SE(3)$ equivariance of the message-passing framework.*

*Proof in Appendix B.2.*

*Remark* 1. The loss of reflection equivariance $E(3) \rightarrow SE(3)$ occurs at the input level through ESM embeddings, not through the attention mechanism itself.

*Remark* 2 (Parameter and Computational Efficiency). The Gaussian Dynamic Attention requires only $O(H)$ learnable parameters complexity, in contrast to the $O(d^2)$ for standard dot-product attention's key-query projection. Calculating neighborhood statistics incurs merely $O(|N(i)|d)$ operations per node, which is insignificant relative to the $O(|N(i)|d^2)$ for conventional dot-product attention. This results in $O(d)$-fold reduction in computational complexity with negligible parameter overhead and a significant decrease in inference time, as shown in Appendix Table 3.

### 3.6 TRAINING OBJECTIVE

We formulate binding site prediction as a multi-task learning problem that jointly optimizes localization accuracy and geometric understanding of protein-ligand interactions.

**Protein Ligand Binding Site Prediction.** For binding site identification we use Dice Loss $\mathcal{L}_{\text{Dice}}$ following (Aggarwal et al., 2021; Zhang et al., 2024) to address inherent class imbalance, then compute $\hat{y}_i = \mathbf{Sigmoid}(MLP(h_i^{(L)})$ are predicted binding probabilities after $L$ layers, $y_i \in \{0, 1\}$ are ground truth labels, and $\epsilon$ is stability factor. The Dice coefficient naturally handles imbalanced classes by focusing on the overlap between predictions and ground truth rather than individual classification accuracy.

**Auxiliary Directional Loss.** To enhance geometric understanding, we extract directional information from the learned equivariant features and supervise it with ground truth ligand directions. The $l = 1$ steerable features $\tilde{\mathbf{X}}_i^{(1)} \in \mathbb{R}^{3 \times h_d}$ inherently encode directional information. We extract predicted directions as: $\hat{\mathbf{d}}_i = \frac{\bar{\mathbf{X}}_{ichannel}^{(1)}}{\| \bar{\mathbf{X}}_{ichannel}^{(1)} \|_2 + \epsilon}$. Here, $\bar{\mathbf{X}}_{ichannel}^{(1)} = \frac{1}{h_d} \sum_{k=1}^{h_d} \tilde{\mathbf{X}}_{i,k}^{(1)} \in \mathbb{R}^3$ averages across feature channels to obtain a single direction vector. The ground truth direction $\mathbf{d}_i^{\text{true}}$ points from residue $i$ to the nearest ligand heavy atom: $\mathbf{d}_i^{\text{true}} = \frac{\mathbf{p}_{\text{lig}}^* - \mathbf{p}_i}{\|\mathbf{p}_{\text{lig}}^* - \mathbf{p}_i\|_2}$, where $\mathbf{p}_{\text{lig}}^* = \arg\min_{\mathbf{p} \in \mathcal{L}} \|\mathbf{p} - \mathbf{p}_i\|_2$. Here, $\mathcal{L}$ denotes the set of ligand atom positions. We compute directional loss $\mathcal{L}_{\text{Dir}}$ using cosine similarity between true and predicted directions. Hence, the training objective of our GDEGAN becomes $\mathcal{L} = \mathcal{L}_{\text{Dice}} + \mathcal{L}_{\text{Dir}}$.

## 4 EXPERIMENTS

### 4.1 DATASETS AND BASELINE METHODS COMPARED

We utilize the datasets benchmark settings of EquiPocket (Zhang et al., 2024) for LBS identification. **scPDB** (Desaphy et al., 2015) is the widely used dataset for training and validation, which contains the proteins' 3D structures and ligands. **PDBbind2020** (Wang et al., 2004), **COACH420** and **HOLO4K** Krivák & Hoksza (2018) are three diverse datasets used to evaluate our method. The detailed discussion on datasets can be found in Appendix C.1. We compare GDEGAN with several categories of methods, as seen in Table 1, with further information provided in Appendix C.2.

### 4.2 EVALUATION METRICS

We used well established metrics **DCC**, **DCA** and **Failure rate** (Chen et al., 2011) for LBS identification, detailed in Appendix C.3. We assess localization accuracy through **DCC** (Distance from Center of Center), measuring the Euclidean distance between predicted and true binding site centers, **DCA** (Distance to Closest Atom), measuring the minimum distance from the predicted center to any ligand atom, and **Failure rate** as percentage of proteins without any predicted binding site. Predictions are considered successful when DCC or DCA falls below the standard 4Å threshold, which

Table 1: Experimental results of baseline models and our framework measured by DCC and DCA success rates[a]. The table presents comparative results across three benchmark datasets. Bold values indicate the best performance in each metric.

| Methods | Param (M) | Failure rate ↓ | COACH420 | | HOLO4K[k] | | PDBbind2020 | |
| --- | --- | --- | --- | --- | --- | --- | --- | --- |
| | | | DCC↑ | DCA↑ | DCC↑ | DCA↑ | DCC↑ | DCA↑ |
| Fpocket[b] | \ | **0.000** | 0.228 | 0.444 | 0.192 | 0.457 | 0.253 | 0.371 |
| P2rank[b] | \ | **0.000** | 0.366 | 0.628 | 0.314 | 0.621 | 0.503 | 0.677 |
| DeepSite[b] | 1.00 | \ | \ | 0.564 | \ | 0.456 | \ | \ |
| Kalasanty[b] | 70.6 | 0.120 | 0.335 | 0.636 | 0.244 | 0.515 | 0.416 | 0.625 |
| DeepSurf[b] | 33.1 | 0.054 | 0.386 | 0.658 | 0.289 | 0.635 | 0.510 | 0.708 |
| RecurPocket[b] | 21.2 | 0.075 | 0.354 | 0.593 | 0.277 | 0.616 | 0.492 | 0.663 |
| GAT[b] | **0.03** | 0.110 | 0.039(0.005) | 0.130(0.009) | 0.036(0.003) | 0.110(0.010) | 0.018(0.001) | 0.088(0.011) |
| GCN[b] | 0.06 | 0.163 | 0.049(0.001) | 0.139(0.010) | 0.044(0.003) | 0.174(0.003) | 0.018(0.001) | 0.070(0.002) |
| GCN2[b] | 0.11 | 0.466 | 0.042(0.098) | 0.131(0.017) | 0.051(0.004) | 0.163(0.008) | 0.023(0.007) | 0.089(0.013) |
| SchNet[b] | 0.49 | 0.140 | 0.168(0.019) | 0.444(0.020) | 0.192(0.005) | 0.501(0.004) | 0.263(0.003) | 0.457(0.004) |
| Egnn[b] | 0.41 | 0.270 | 0.156(0.017) | 0.361(0.020) | 0.127(0.005) | 0.406(0.004) | 0.143(0.007) | 0.302(0.006) |
| EquiPocket[b] | 1.70 | 0.051 | 0.423(0.014) | 0.656(0.007) | 0.337(0.006) | 0.662(0.007) | 0.545(0.010) | 0.721(0.004) |
| GotenNet | 2.20 | 0.049 | 0.464(0.007) | 0.624(0.014) | 0.454(0.001) | 0.691(0.005) | 0.553(0.008) | 0.705(0.007) |
| **GDEGAN** | 1.90 | 0.032 | **0.580(0.008)** | **0.707(0.009)** | **0.560(0.013)** | **0.788(0.011)** | **0.675(0.010)** | **0.826(0.011)** |

[1] [a]The standard deviation is indicated in brackets. [b] Results from the EquiPocket (Zhang et al., 2024) paper. [k] holo4k contains multi chains and complex with multiple copies, presenting a strong distribution shift.

Table 2: Ablation Study.

| Methods | EQ[c] | ADL[d] | ESM[e] | COACH420 | | HOLO4K | | PDBbind2020 | |
| --- | --- | --- | --- | --- | --- | --- | --- | --- | --- |
| | | | | DCC↑ | DCA↑ | DCC↑ | DCA↑ | DCC↑ | DCA↑ |
| GotenNet | $E(3)$ | No | No | 0.454(0.007) | 0.624(0.014) | 0.464(0.001) | 0.691(0.005) | 0.553(0.008) | 0.705(0.007) |
| GotenNet+ADL | $E(3)$ | Yes | No | 0.485(0.004) | 0.642(0.011) | 0.468(0.004) | 0.732(0.004) | 0.592(0.010) | 0.748(0.003) |
| GotenNet+ESM | $SE(3)$ | No | Yes | 0.543(0.008) | 0.693(0.006) | 0.520(0.011) | 0.753(0.004) | 0.637(0.005) | 0.760(0.004) |
| GotenNet(full) | $SE(3)$ | Yes | Yes | 0.556(0.002) | 0.703(0.005) | 0.529(0.011) | 0.749(0.006) | 0.649(0.005) | 0.801(0.014) |
| **GDEGAN+ESM** | $SE(3)$ | No | Yes | 0.572(0.001) | 0.702(0.002) | 0.532(0.010) | 0.769(0.006) | 0.652(0.010) | 0.810(0.011) |
| **GDEGAN(full)** | $SE(3)$ | Yes | Yes | **0.580(0.008)** | **0.707(0.009)** | **0.560(0.013)** | **0.788(0.011)** | **0.675(0.010)** | **0.826(0.011)** |

[1] [c]Equivariance of the Model. [d] Auxiliary Directional Loss. [e] Node Features generated through ESM-2 (Lin et al., 2022). GotenNet computes dot-product attention while GDEGAN computes gaussian dynamic attention.

captures typical protein-ligand interaction distances. During inference on novel proteins where the number of binding pockets are unknown, we employ mean-shift clustering (Comaniciu & Meer, 2002) on high-scoring residues ($\hat{y}_i > \tau$) following (Krivák & Hoksza, 2018) to automatically identify multiple binding pockets.

## 4.3 IMPLEMENTATION DETAILS

We used 4 layers of GDEGAN with hidden dimensions as 128 throughout the model, $L_{max} = 2$ for steerable features, attention heads as 8, and edge spatial cutoff $r_c$ set to 10Å. We trained our model using the AdamW Optimizer (Loshchilov & Hutter, 2019) for 100 epochs, selecting the best checkpoints based on the validation set. LayerNorm, TensorLayerNorm and Dropouts (Aykent & Xia, 2025) were applied in each layer with SiLU activation. The learning rate was initially set to 0.0005 with Cosine Scheduler and weight decay with value 0.05. We trained our model on NVIDIA H100 NVL GPU with a batch size of 16. All the hyperparameters were selected based on the validation dataset, which is the 10% of the training dataset. More details are provided in Appendix F.

## 4.4 RESULTS AND ATTENTION PATTERS ANALYSIS

The experimental results shown in Table 1 indicate substantial improvements across all benchmarks. GDEGAN attains the highest DCC success rate across all test datasets, with significant enhancements of 37.1%, 66.17%, and 23.8% over EquiPocket's DCC success rate for COACH420, HOLO4k, and PDBbind2020, respectively. For DCA metrics, we achieve enhancements of 7.7%, 19.0%, and 14.5% compared Equipocket's DCA. GDEGAN decreases the failure rate to 3.2%, in

contrast to 5.1% for EquiPocket and 4.9% for GotenNet. Figure 2 illustrates the visualization of both predictions: the centroid and probable binding site candidates from our model. The comparison of inference speeds for our method is presented in Appendix C.4.

We visualize learned attention patterns for both Gaussian dynamic attention (GDEGAN) and dot product attention (GotenNet) as shown in Figure 3. The binding site attention sub-matrix on the left of GDEGAN shows higher values on binding site residues (darker red) and relevant neighbors (dark yellow), suggesting strong local clustering, which demonstrates the adaptive nature of the method. We quantitatively validate this adaptive behavior by our variance analysis (Appendix E), where we demonstrate that binding sites exhibit significantly heigher learned variance ($\sigma_i^2 = 1.41 \pm 0.52$) compared to non-binding regions ($\sigma_i^2 = 0.75 \pm 0.35$, $p < 0.001$). This statistical validation directly supports our approach, aligning with the claim

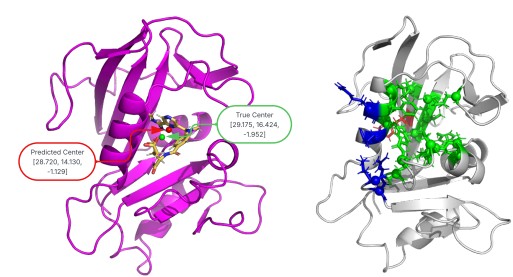

Figure 2: **Visualization of Protein 'PDB:1u72(A)'.** **Left:** Model prediction (red) vs true center (green) with coordinates. **Right:** Predicted residues: True Positive (green), False Positive (red), False Negative (blue).

that GDEGAN's Gaussian dynamic attention impose localization by automatically nullifying distant interactions. This crucial localization allows the adaptive nature of GDA to effectively discriminate between binding pockets and surface regions by concentrating the model's representational capability on statistically relevant neighborhoods. Meanwhile, the attention sub-matrix on the left of GotenNet shows a typical pattern of dot product attention capturing binding sites, but with less concentration on relevant neighbors. The attention values with reduced peak magnitudes in GDEGAN indicate optimal allocation of attention instead of indiscriminate distribution.

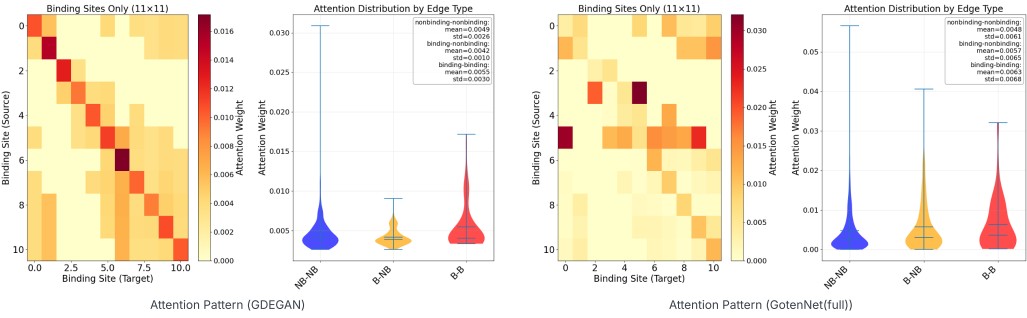

Figure 3: **Attention Patters Visualizations of Protein 'PDB:3c2f(A)'. Left:** On the left we show the attention patterns of GDEGAN, and on the **Right:** attention patterns of GotenNet(full).

## 4.5 ABLATION STUDY

Table 2 presents the component-wise ablation results. Using ESM embeddings (GotenNet+ESM) instead of atomic numbers embeddings (GotenNet) improves the results by 15.61% DCC and 9.27% DCA averaged across datasets, indicating evolutionary information helps capture better binding regions than a purely geometric approach. Gaussian attention (GDEGAN+ESM) improves the results over dot-product attention (GotenNet+ESM) by 3.33% DCC and 3.32% DCA and both the techniques (full) with auxiliary directional loss as directional supervision further improves the results by an average of 2% DCC and 3.5% DCA respectively averaged across datasets. Critically, on the more structurally diverse HOLO4K dataset, GDEGAN(full) achieves a 2.4% DCC improvement over GotenNet(full), compared to only 1.5% on the more homogeneous COACH420 dataset, demonstrating that statistical adaptation provides greater benefits as structural heterogeneity increases. Notably, ESM features provide the largest enhancement, Gaussian attention's contribution grows with data complexity, validating our hypothesis that adaptive statistical kernels better handle protein heterogeneity than uniform similarity measures. Figure 4a shows the relationship between model depth, parameter count, and performance. We observe that performance peaks at 4 layers (average DCC =

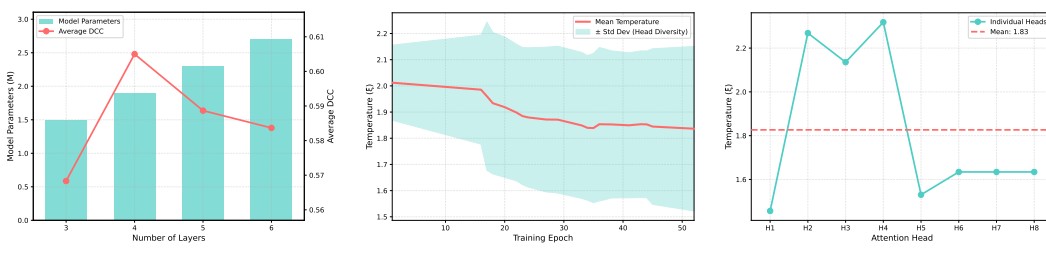

(a) Model Depth vs. Performance     (b) Temperature Evolution     (c) Final Temperature Distribution

Figure 4: **Ablation analysis.** **(a)** Model depth analysis showing performance peaks at 4 layers with increased parameters leading to oversmoothing. **(b)** Temperature evolution across training epochs averaged across multiple runs with mean and standard deviation. **(c)** Learned temperatures distribution across 8 attention heads (best model).

0.605) and degrades with additional layers despite increased parameters (6 layers: 2.70M parameters, average DCC = 0.584). EquiPocket (Zhang et al., 2024) performance peaks at 3 layers where as GotenNet performance peaks at 4 layers, both of the methods are do-product attention based methods. This occurs because dot-product attention computation is fundamentally pairwise, computing similarity between node pairs independently, while variance requires neighborhood-level aggregation. Our GDA explicitly computes $\sigma_i^2$ at each layer, providing direct inductive bias. Furthermore, our variance analysis (Appendix E) shows binding sites have $1.89\times$ higher variance ($r = 0.39$, $p < 0.001$, $d = 1.85$), confirming explicit computation captures biologically meaningful signals that standard attention cannot learn implicitly. The learnable temperatures distribution during training for multiple runs (mean: $2.03 \rightarrow 1.826$, Figure 4b with heads specializing into low-$\xi$ ($\approx 1.4561$, selective) and high-$\xi$ ($\approx 2.3181$, inclusive) groups (Figure 4c), increasing diversity $5.6\times$ (std: $0.059 \rightarrow 0.330$). This multi-scale specialization, enables adaptive pattern recognition for diverse binding site geometries.

## 5   CONCLUSION AND LIMITATIONS

This study presents GDEGAN, an enhancement of GotenNet (Aykent & Xia, 2025), which substitutes dot-product attention with Gaussian Dynamic Attention, specifically developed for the detection of protein-ligand interaction sites. This simple yet efficacious change demonstrates enhancements of 42.36% and 13.73% in DCC and DCA, respectively, averaged across all datasets in comparison to the prevailing state-of-the-art approach, EquiPocket (Zhang et al., 2024). Unlike previous methods that utilize atom-level information (Jiménez et al., 2017; Zhang et al., 2024; Aggarwal et al., 2021), our approach depends on residue-level information, hence improving efficiency in both training and inference due to a reduction in input graph size.

Our method is proposed to find the binding center by predicting probable binding site candidates instead of performing docking, a logical progression is to utilize our predictions to limit the docking search space. Even though the training data is limited in size and data points have well-defined pockets with a single ligand, our approach can generalize better. Our method would benefit more from training on more diverse data points with multi-ligand interacting pockets. Given that GDEGAN predicts binding sites Future research should methodically assess whether our statistical attention scores indicate ligand binding locations and predict binding affinity based on local feature coherence.

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

## A  GEOMETRIC GRAPH FORMULATION AND BINDING SITE REPRESENTATIONS

### A.1  REPRESENTATION OF PROTEINS

The 3D structure of a protein is defined by the spatial coordinates of atoms associated with every amino acid, organized according to its amino acid sequence. For computational efficiency and biological relevance in detecting probable binding site candidates, we represent each amino acid residue by its $C_\alpha$ atom position $\mathbf{p}_i \in \mathbb{R}^3$, which provides a stable backbone reference point consistent across all amino acid types. We formalize a protein structure as a geometric graph $\mathcal{G} = (\mathcal{V}, \mathcal{E})$ that captures both topological and spatial information necessary for binding site identification. Nodes in the geometric graph $\mathcal{V} = \{(\mathbf{p}_i, h_i)\}_{i=1}^N$ represent residue-feature pairs. The edges are established from each residue $i$ to all residues $j$ within a cutoff radius: $\mathcal{E} = \{(i, j) : \|\mathbf{p}_i - \mathbf{p}_j\| < r_c, i \neq j\}$, where we set $r_c = 10$ Å based on typical interaction distances in protein binding sites following (Zhang et al., 2024). Rather than encoding raw physicochemical properties, we leverage learned representations that capture both evolutionary conservation and structural context. Specifically, for each residue $i$, we extract a feature vector $h_i \in \mathbb{R}^d$ from the pretrained ESM-2 (Lin et al., 2022) model, which encodes evolutionary information and sequence context.

### A.2  BINDING SITE DEFINITION AND REPRESENTATION.

Protein binding sites are the critical regions where ligands, or other molecules interact with proteins in the biological process. We define binding sites based on proximity to known ligand positions from crystal structures, surrounded by the atoms of the protein. A residue is classified as belonging to the binding site when any of its constituent atoms lies within a threshold distance $d_{bind}$ from any ligand atom. Formally, a residue $i$ is considered part of a binding site as:

$$y_i = \begin{cases} 1 & \text{if } \min_{a \in \mathcal{A}_i, b \in \mathcal{L}} \|\mathbf{p}_a - \mathbf{p}_b\| < d_{bind} \\ 0 & \text{otherwise} \end{cases} \tag{16}$$

where $\mathcal{A}_i$ represents all atoms of residue $i$, $\mathcal{L}$ represents all ligand atoms. $d_{bind}$ represents the binding distance threshold taken as 4Å  following (Zhang et al., 2024; Krivák & Hoksza, 2018; Jiménez et al., 2017).

## B  PROOFS

### B.1  PROOF OF PROPOSITION 3.1

*Proof.* Considering the GDEGAN architecture with ESM embeddings as node features. We show that the network maintains SE(3) but not E(3) equivariance.

**ESM embeddings are invariant scalars.** ESM embeddings $h_i \in \mathbb{R}^{n_d}$ encode amino acid sequence information and evolutionary patterns. These are scalar features that do not transform under any spatial transformation. For any transformation $g = (\mathbf{R}, \mathbf{t})$ where $\mathbf{R} \in SO(3)$ (rotation) and $\mathbf{t} \in \mathbb{R}^3$ (translation):

$$h'_i = h_i \quad \forall g \in SE(3) \tag{17}$$

**Loss of reflection equivariance.** Consider a chiral molecule $\mathcal{M}$ and its mirror image $\mathcal{M}'$ obtained by reflection $P$. Both have identical ESM embeddings: $h_i = h'_i$ (same amino acid sequence) and identical pairwise distances: $\|\mathbf{x}_i - \mathbf{x}_j\| = \|\mathbf{x}'_i - \mathbf{x}'_j\|$, but different chirality. Since the network

cannot distinguish between $\mathcal{M}$ and $\mathcal{M}'$ based on invariant features alone, it cannot be equivariant under reflections.

$\square$

## B.2 PROOF OF GAUSSIAN DYNAMIC ATTENTION EQUIVARIANCE 3.2

Here we prove that our Gaussian Dynamic Attention mechanism maintains SE(3) equivariance:

### B.2.1 GAUSSIAN ATTENTION MECHANISM

Our Gaussian attention computes attention weights as:

$$d_{ij}^{\text{scaled}} = \frac{h_j - h_i}{\sqrt{\sigma_i^2 + \epsilon}} \tag{18}$$

$$\alpha_{ij} = \frac{\exp\left(-\frac{(\|d_{ij}^{\text{scaled}}\|_2)^2}{2\xi}\right)}{\sum_{k \in \mathcal{N}(i)} \exp\left(-\frac{(\|d_{ik}^{\text{scaled}}\|_2)^2}{2\xi}\right)} \tag{19}$$

where $h_i, h_j \in \mathbb{R}^{n_d}$ are invariant scalar features from ESM embeddings, $\sigma_i^2$ is the neighborhood variance, and $\xi$ is the learnable temperature.

### B.2.2 PROOF OF SE(3) EQUIVARIANCE

*Proof.* **Invariance of scalar features.** The ESM embeddings $h_i$ are invariant under SE(3) transformations, because they encode sequential and evolutionary information, not geometric co-ordinates:

$$g(h_i) = h_i \tag{20}$$

**Invariance of attention weights.** Since $h_i$ and $h_j$ are invariant:

$$\|g(h_j) - g(h_i)\|^2 = \|h_j - h_i\|^2 \tag{21}$$

The neighborhood variance $\sigma_i^2$ computed from invariant features is also invariant:

$$\sigma_i^2 = \frac{1}{|\mathcal{N}(i)|} \sum_{k \in \mathcal{N}(i)} \|h_k - \mu_i\|^2 \tag{22}$$

where $\mu_i$ is the neighborhood mean. Under $g \in SE(3)$, both remain unchanged.

Therefore:

$$\alpha_{ij}^g = \exp\left(-\frac{(\|d_{ij}^{\text{scaled}}\|_2)^2}{2\xi}\right) = \alpha_{ij} \tag{23}$$

Hence, Gaussian Dynamic Attention mechanism preserves SE(3) equivariance by maintaining invariant attention weights while allowing equivariant features to transform appropriately under rotations. The scalar nature of ESM embeddings ensures that reflection equivariance is not required, reducing $E(3)$ to $SE(3)$ as stated in Proposition 3.1. $\square$

## C EXPERIMENTS

### C.1 DATASETS

**scPDB** (Desaphy et al., 2015) comprises 17,594 protein-ligand complex structures from the 2017 release, representing 4,782 unique proteins and 6,326 ligands. We used the most frequently used dataset for LBS identification (Stepniewska-Dziubinska et al., 2020; Jeevan et al., 2024) for training and validation, with a split of $90 : 10$. Final dataset was preprocessed using the steps described in EquiPocket (Zhang et al., 2024). **PDBbind2020** Wang et al. (2004) contains experimentally determined binding affinity data paired with structural information. We utilize the refined subset as per

(Zhang et al., 2024), consisting of 5,316 complexes selected for structural quality from the larger general set of 14,127 complexes. The refined set enforces strict quality criteria, including resolution better than 2.5Å and complete ligand electron density. **COACH420 and HOLO4K** serve as independent test sets following Krivák & Hoksza (2018). COACH420 contains 420 protein-ligand complexes with diverse binding site architectures, while HOLO4K comprises 4,288 structures. Both datasets use the $MLIG$ subsets following (Aggarwal et al., 2021; Jiménez et al., 2017) containing biologically relevant ligands as defined by the original curation. Notably, HOLO4K presents significant distribution shift challenges as it contains numerous multi-chain assemblies and oligomeric proteins absent from typical training sets. We have split the HOLO4K dataset into per-chain components and aggregated the predictions in our evaluated results. For all datasets, we exclude solvent molecules and apply standard preprocessing, such as removing hydrogen atoms. Structures with missing coordinates or ambiguous ligand positions are filtered during preprocessing using rDkit (Tosco et al., 2014).

## C.2 BASELINE METHODS COMPARED

We compare GDEGAN with different categories of methods proposed for LBS identification. *Traditional Machine Learning-based:* Fpocket (Le Guilloux et al., 2009), and P2rank (Krivák & Hoksza, 2018). *CNN-based:* DeepSite (Aggarwal et al., 2021), Kalasanty (Stepniewska-Dziubinska et al., 2020), and RecurPocket (Li et al., 2022). *Topological Graph-based:* GAT Veličković et al. (2018), GCN Kipf & Welling (2017), and GCN2 (Chen et al., 2020). *Spatial Graph-based:* SchNet (Schütt et al., 2018), EGNN (Satorras et al., 2021), and Equipocket (Zhang et al., 2024). *High-degree steerable method:* GotenNet (Aykent & Xia, 2025).

## C.3 EVALUATION METRICS

**DCC (Distance from Center to Center).** For each predicted binding site center $\hat{\mathbf{p}}_i$ and true binding site center $\mathbf{p}_j$, DCC measures the Euclidean distance between centers:

$$\text{DCC} = \|\hat{\mathbf{p}}_i - \mathbf{p}_{ligand}\|_2 \tag{24}$$

where $\hat{\mathbf{p}}_i \in \mathbb{R}^3$ represents the $i$-th predicted center and $\mathbf{p}_j \in \mathbb{R}^3$ the $j$-th ground truth center.

**DCA(Distance to Closest Atom).** This metric evaluates whether predictions are within the actual binding region by measuring the minimum Euclidean distance from a predicted center to any ligand atom:

$$\text{DCA} = \min_{b \in \mathcal{L}} \|\hat{\mathbf{p}}_i - \mathbf{p}_b\|_2 \tag{25}$$

where $\mathcal{L}$ represents all ligand atoms.

For both metrics predictions are considered successful if they are within a standard threshold $\tau$, which in this case we have taken as 4Å following Aggarwal et al. (2021); Le Guilloux et al. (2009); Mylonas et al. (2021); Zhang et al. (2024).

$$\text{Success Rate}_{\text{DCC/DCA}} = \frac{|\{\text{Predicted sites} \mid \text{DCC/DCA} < \tau\}|}{|\{\text{True sites}\}|} \tag{26}$$

$$\text{Failure Rate} = \frac{|\{\text{Proteins} \mid |\text{predicted centers}| = 0\}|}{|\{\text{Proteins}\}|} \tag{27}$$

where $|\cdot|$ denotes set cardinality and $\tau = 4$Å is the standard threshold for successful prediction.

We have used **DCC/DCA success rate** and **Failure rate** as the evaluation metrics to compare from the sate-of-the-art methods.

## C.4 COMPUTATIONAL EFFICIENCY ANALYSIS

We evaluate the inference speed on 100 proteins for each model and present the duration in seconds (s) in Table 3. GDEGAN demonstrates substantial enhancements, with inference time about 1.9 seconds, in contrast to GotenNet's 4.12 seconds and Equipocket's 37 seconds.

Table 3: Inference time comparison across methods.

| Method | Time (s/100 proteins) | Speedup | Type |
|---|---|---|---|
| **GDEGAN (Ours)** | **1.90** | **19.5×** | Reside Level Nodes |
| GotenNet | 4.12 | 9.0× | Reside Level Nodes |
| EquiPocket[p] | 37.00 | 1.0× | Atom Level Nodes |
| Fpocket[p] | 23.00 | 1.6× | Geometry Based |
| Kalasanty[p] | 86.00 | 0.4× | 3D-CNN Based |
| DeepSurf[p] | 641.00 | 0.06× | 3D-CNN Based |

[p] Results from the EquiPocket (Zhang et al., 2024) paper.

## D    TRAINING STABILITY ANALYSIS

We have conducted a comprehensive stability analysis comparing GDEGAN and GotelNet(full) across training epochs until training converges from early-stopping criteria. The empirical evidence from the training and validation loss shows that $(\sigma_i)^2$ term in our GDEGAN not only maintains the stability but also improves training convergence compared to GotenNet(full). GDEGAN demonstrates **5.3% lower training loss variance (.0179 vs .0189)** compared to GotenNet(full), indicating more stable gradient flow despite $(\sigma_i)^2$ computation. Throughout training we observed monotonic convergence without divergence or oscillations. Figure 5 shows training and validation plots until 50 epochs consistent with our early stopping criterion that selects the best model. Interestingly, while GDEGAN slightly higher validation loss, it achieves better test performance as indicated in Table 2. This validation-test gap indicates that our method learns the right features for binding site localization rather than overfitting to loss minimization.

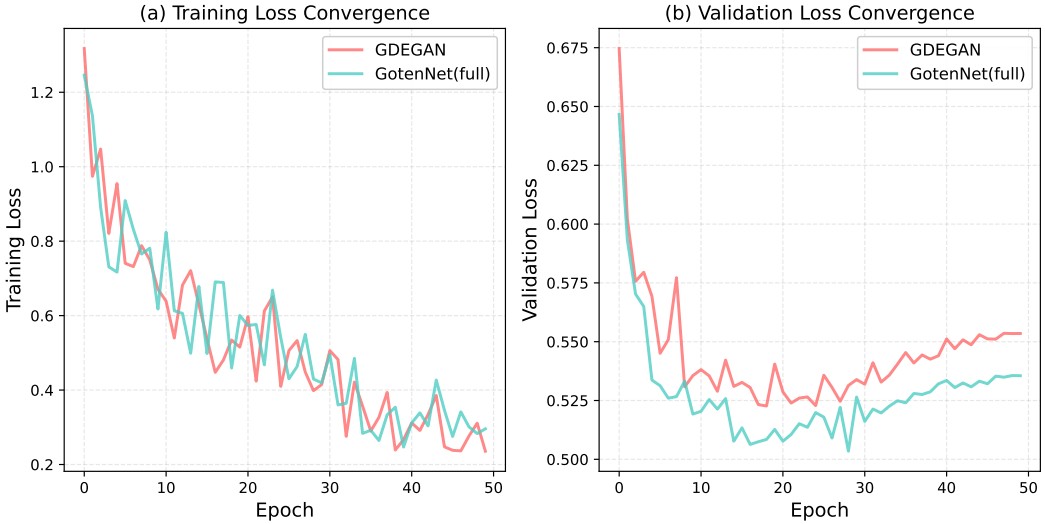

Figure 5: Training and Validation loss curve. **Left:** On the left (a) we show training loss convergence, and on the **Right:** (b) validation loss convergence.

## E    DEPENDENCY BETWEEN VARIANCE AND BINDING SITES

We hypothesize that the binding site exhibits high variance in local structure. To validate our hypothesis we conducted a comprehensive dependency analysis between feature variance and binding sites on the validation set. This analysis directly evaluates whether the learned neighborhood variance $(\sigma_i^2)$ computed by our Gaussian Dynamic Attention mechanism correlates with ground-truth binding site locations. We extracted the learned neighborhood variance $\sigma_i^2$ from the final Gaussian dynamic attention layer of our trained GDEGAN model on the validation set, comprising 205,791 residues (11,107 binding sites, 194,684 non-binding residues). The validation set represents 10% of

the scPDB training data, ensuring independent evaluation. We conducted multiple statistical tests to comprehensively evaluate the variance-binding relationship.

Table 4: Statistical Analysis of Variance-Binding Site Correlation

| Metric | Value | p-value |
|---|---|---|
| *Correlation Coefficients* | | |
| Pearson correlation ($r$) | 0.3865 | <0.001 |
| Spearman correlation ($\rho$) | 0.2900 | <0.001 |
| Point-biserial correlation | 0.3865 | <0.001 |
| *Hypothesis Tests* | | |
| Mann-Whitney U statistic | $1.88 \times 10^9$ | <0.001 |
| Independent t-test ($t$) | 190.10 | <0.001 |
| *Effect Size* | | |
| Cohen's $d$ | 1.8545 | – |
| *Descriptive Statistics* | | |
| Binding residues (mean $\pm$ SD) | $1.412 \pm 0.522$ | – |
| Non-binding residues (mean $\pm$ SD) | $0.747 \pm 0.347$ | – |
| Ratio (binding/non-binding) | $1.89\times$ | – |

Multiple correlation tests presented in Table 4 and a box-plot in Figure 6 confirm a strong positive relationship between variance and binding sites: Pearson correlation $r = 0.3865$ ($p < 0.001$) for linear association, Spearman $\rho = 0.2900$ ($p < 0.001$) for monotonic relationships, and point-biserial $r = 0.3865$ ($p < 0.001$) for continuous-binary pairs. The very large effect size (Cohen's $d = 1.8545$, far exceeding the conventional threshold of $d = 0.8$ for "large") indicates this difference is not only statistically significant but also practically meaningful, demonstrating that variance provides strong discriminative power for binding site identification.

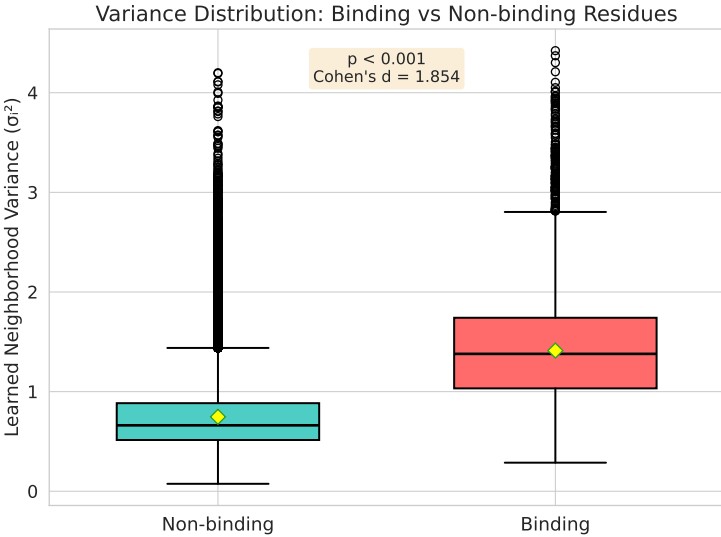

Figure 6: **Variance Distribution: Binding vs Non-binding Residues.** Box plot comparing learned neighborhood variance ($\sigma_i^2$) between binding (red, $n = 11{,}107$) and non-binding (blue, $n = 194{,}684$) residues on the validation set.

## F  TRAINING HYPER-PARAMETERS SELECTION

This section presents the hyperparameters for training as outlined in Table 5, selected based on the validation data, which comprises 10% of the training dataset. These hyperparameters can be

employed to ensure reproducibility. **We will release the full code based on the acceptance of the work.**

Table 5: Hyperparameter selection and reproducibility details.

| Hyperparameter | Search Space |
|---|---|
| Learning Rate | {0.003, 0.0003, **0.0005**} |
| Minimum Learning Rate | **1e-6** |
| Batch Size | {8, **16**, 32} |
| Optimizer | {Adam, **AdamW**} |
| Learning rate scheduler | **Cosine Annealing Warm Restarts** |
| Warmup Epochs | **10** |
| Maximum Epochs | **100** |
| Early Stopping Patience | **30** |
| Gradient clipping | {10, **15** } |
| Weight Decay | {0.01, **0.05**} |
| Dropout Rate | {0.1, 0.2, **0.5**} |
| Node hidden dimension | **128** |
| Edge dimension ($e_d$) | **128** |
| Edge refinement dimension | **128** |
| $L_{max}$ | **2** |
| Number of Layers | {3, **4**, 5, 6} |
| Number of RBFs | **32** |
| Maximum number of neighbors | **32** |
| Number of attention heads | {4, **8**} |
| Activation Function | {ReLU, **SiLU**} |
| $\tau$ | **0.5** |

## G  HYPERPARAMETER SENSITIVITY ANALYSIS

In Table 6 we present the performance of GDEGAN for different message passing layer, to perform hyperparameter sensitivity analysis. Consistent with prior observations in equivariant GNNs (Zhang et al., 2024), we find that excessive message passing leads to oversmoothing—neighboring nodes become increasingly similar, losing discriminative power. For GDEGAN, performance peaks at 4 layers (Average DCC=0.6050) and degrades with deeper architectures (5 layers: 0.588, 6 layers: 0.583), despite each additional layer adding nearly 20% more parameters.

Table 6: Performance of GDEGAN for varying number of message passing layers. All the other parameters were kept same to ensure fare comparison.

| layers | Param (M) | COACH420 | | HOLO4K | | PDBbind2020 | |
|---|---|---|---|---|---|---|---|
| | | DCC↑ | DCA↑ | DCC↑ | DCA↑ | DCC↑ | DCA↑ |
| 3 | 1.50 | 0.528(0.002) | 0.673(0.010) | 0.534(0.002) | 0.731(0.009) | 0.643(0.005) | 0.777(0.002) |
| **4 (default)** | **1.90** | **0.580(0.008)** | **0.707(0.009)** | **0.560(0.013)** | **0.788(0.011)** | **0.675(0.010)** | **0.826(0.011)** |
| 5 | 2.30 | 0.564(0.002) | 0.680(0.010) | 0.544(0.003) | 0.751(0.008) | 0.658(0.004) | 0.800(0.010) |
| 6 | 2.70 | 0.550(0.005) | 0.681(0.004) | 0.541(0.002) | 0.749(0.012) | 0.660(0.010) | 0.800(0.017) |

## H  DECLARATION ON THE USE OF LARGE LANGUAGE MODELS

In this work, we have utilised tools like **Grammarly** to check any grammatical oversight, and these tools are powered by LLMs.

