# OpenReview forum: "GDEGAN: Gaussian Dynamic Equivariant Graph Attention Network for Ligand Binding Site Prediction"
_ICLR.cc/2026/Conference — ICLR 2026 Conference Withdrawn Submission_

### Official Review · Reviewer_TS28 · 2025-10-22

**Soundness:** 2
**Presentation:** 2
**Contribution:** 3
**Rating:** 4
**Confidence:** 2

**Summary:**

This paper introduces GDEGAN, a novel Gaussian Dynamic Equivariant Graph Attention Network designed for predicting protein-ligand binding sites. The core innovation is the replacement of standard dot-product attention with a Gaussian Dynamic Attention mechanism. This new mechanism adapts to the local chemical and geometric heterogeneity of protein surfaces by using the mean and variance of neighboring residue features to compute attention scores, leading to state-of-the-art performance on three established benchmark datasets.

**Strengths:**

The novelty lies in the successful adaptation and application of a probabilistic, variance-aware attention mechanism to the domain of 3D equivariant graph representations for a critical bioinformatics task. While building upon a strong backbone (GotenNet), the proposed attention module is a distinct and impactful innovation. It provides a more physically grounded inductive bias by assuming that variance in learned features is a meaningful signal, a departure from standard similarity-based dot-product attention.

The model achieves substantial and consistent improvements over strong baselines, including the current state-of-the-art EquiPocket, across three diverse datasets (COACH420, HOLO4k, PDBbind2020). The reported relative gains (e.g., 37-66% in DCC) are compelling.

**Weaknesses:**

**Key Flaw:** The most critical weakness is the ambiguity in the formulation of the core Gaussian attention mechanism. Specifically, the dimensionality of the neighborhood statistics μ_i and (σ_i)^2 in Equations 5 and 6 is unclear in the context of Equation 7. Since h_j is a high-dimensional feature vector, μ_i and (σ_i)^2 should also be vectors (element-wise mean and variance). However, Equation 7 uses (σ_i)^2 as if it were a scalar value for modulating the attention kernel's bandwidth. This lack of clarity is a major impediment to understanding and reproducing the method and casts doubt on its technical soundness. The authors must explicitly define how the vector variance is converted into the scalar used in the denominator.

**Methodological Issues:** The central hypothesis of the paper, while intuitive, could be better substantiated. The authors assume that high local feature variance is a reliable signal for binding sites and that standard dot-product attention cannot capture this.

1. Justification of Hypothesis: This assumption is presented as a given but lacks direct empirical or theoretical support. Is there prior work suggesting a strong correlation between feature variance and functional sites?

2. Expressive Power of Baselines: The paper argues that multi-layer GNNs with standard attention are insufficient. However, it is plausible that a sufficiently deep model could learn to approximate similar context-aware behavior implicitly. The paper does not provide a compelling argument or experiment to rule out this possibility.

**Experimental Evaluation Issues:** The experimental section is strong but could be improved.

1. Training Dynamics: The introduction of a data-dependent variance term (σ_i)^2 in the denominator of an exponential function could potentially lead to training instability (e.g., vanishing or exploding gradients) if the variance becomes very small or large. The paper does not discuss the training dynamics or present loss curves to demonstrate that the model converges as robustly as the baseline.

2. Lack of Parameter Sensitivity Analysis: The model introduces a learnable temperature parameter ξ for each attention head. An analysis of the model's sensitivity to this hyperparameter would strengthen the results and provide insights into the mechanism's behavior.

**Questions:**

Clarifying these issues will be crucial for a more thorough assessment of the paper's quality.

1. Clarification of Equations 5-7: This is the most critical point. Please provide a precise mathematical definition for how the neighborhood variance (σ_i)^2 is used in Equation 7. Given that h is a vector, (σ_i)^2 from Equation 6 should also be a vector. How is this vector transformed into the scalar value required in the denominator of Equation 7? Is it the mean of the vector elements, their L2 norm, or something else?

2. Justification of the Core Hypothesis: Could you provide further justification for the core assumption that local feature variance is a superior signal for identifying binding sites compared to what can be learned by standard attention mechanisms? Perhaps you could show a correlation analysis on a validation set between the learned (σ_i)^2 values and ground-truth binding site locations.

3. Training Stability: Did the use of the (σ_i)^2 term in the attention calculation lead to any training instability? Could you please present the training and validation loss curves for GDEGAN and the GotenNet(full) baseline to demonstrate that the proposed mechanism allows for stable convergence?

4. Computational Overhead: Remark 2 discusses the theoretical computational complexity. Could you provide the empirical wall-clock inference and training time overhead of the Gaussian Dynamic Attention layer compared to the standard dot-product attention layer in GotenNet? This would give a clearer picture of the practical trade-offs.

---

> ### Author Response · Authors · 2025-11-20
> **Point-by-point responses to Reviewer TS28 (1/n)**
>
> Dear Reviewer TS28,
>
> We thank you for acknowledging the strengths, novelty of our research and the valuable feedback. We are happy that you found our approach impactful. We sincerely appreciate the detailed comments and questions. Below we Provide the point-by-point response to each of the comments and questions asked.
>
> > **W1-Q1: Mathematical Ambiguity Clarification in Equations**
>
> Our sincere apologies for the notational confusion in Equations 5-7. Here we provide the description of core gaussian dynamic attention mechanism and clarified this in the **revised manuscript section 3.3 equation (5-8)** to ensure reproducibility. Our proposed mechanism computes:
> 1. Element-wise neighborhood mean:
> $\mu_i = \frac{1}{|\mathcal{N}(i)|} \sum_{j \in \mathcal{N}(i)} h_j$
>
> 2. Element-wise neighborhood variance:
> $(\sigma_i)^2 = \frac{1}{|\mathcal{N}(i)|} \sum_{j \in \mathcal{N}(i)} (h_j - \mu_i)^2$
>
> 3. Dimension wise normalization, with stability factor $\epsilon$ :
> $d_{ij}^{\text{scaled}} = \frac{h_j - h_i}{\sqrt{\sigma_i^2 + \epsilon}}$
>
> 4. Gaussian attention scores:
> $\alpha_{ij} = \frac{\exp\left(-\frac{\|d_{ij}^{\text{scaled}}\|^2}{2\xi}\right)}{\sum_{k \in \mathcal{N}(i)} \exp\left(-\frac{\|d_{ik}^{\text{scaled}}\|^2}{2\xi}\right)}$
>
> We use **element-wise** variance for normalization, then compute the **L2 norm** of the normalized differences. This provides: 1. Dimension-specific adaptive scaling based on local feature diversity, 2. Stability through element-wise epsilon addition.
>
> > **W2-Q2: Methodological Issue (Justification of Core Hypothesis)**
>
> We agree with the reviewer on core assumptions of our work are based upon the relationship between feature variance and functional relevance. While we initially introduced the rationale for this assumption in **Section 3.3 (line 237-238)**, we recognized the reviewers suggestions and added more discussion supported by prior literature. Liang et. al. [1] concludes that binding pockets are characterized by high geometric and chemical heterogeneity. P2Rank (cited in our manuscript), a successful classical method, uses local neighborhood descriptors that implicitly capture variance through chemical diversity features. DeepSite (one of our compared benchmark methods), used 3D-CNN, learns local spatial patterns, essentially capturing local heterogeneity.
>
> To address the question regarding the justification for using local feature variance as a signal for binding site identification, we conducted a comprehensive correlation analysis on the validation set. This analysis directly evaluates whether the learned neighborhood variance $(\sigma_i^2)$ in our Gaussian Dynamic Attention mechanism correlates with ground-truth binding site locations. We observe a strong, statistically significant positive correlation (Pearson r = 0.3865, p $<$ 0.001) with a very large effect size (Cohen's d = 1.8545). **Binding site residues exhibit 1.89 times higher learned variance compared to non-binding residues (mean: 1.412 vs 0.747, p $<$ 0.001).** These results provide robust empirical evidence that local feature variance is indeed a superior signal for binding site identification. This validates our hypothesis that binding pockets, characterized by geometric and chemical heterogeneity, manifest as high-variance regions in learned feature space.
>
>
> | **Metric**                                   | **Value**               | **p-value** |
> |----------------------------------------------|--------------------------|-------------|
> | *Correlation Coefficients*                   |                          |             |
> | &nbsp;&nbsp;Pearson correlation (*r*)        | 0.3865                   | <0.001      |
> | &nbsp;&nbsp;Spearman correlation (*ρ*)       | 0.2900                   | <0.001      |
> | &nbsp;&nbsp;Point-biserial correlation       | 0.3865                   | <0.001      |
> | *Hypothesis Tests*                           |                          |             |
> | &nbsp;&nbsp;Mann-Whitney U statistic         | 1.88 × 10⁹               | <0.001      |
> | &nbsp;&nbsp;Independent t-test (*t*)         | 190.10                   | <0.001      |
> | *Effect Size*                                |                          |             |
> | &nbsp;&nbsp;Cohen’s *d*                      | 1.8545                   | --          |
> | *Descriptive Statistics*                     |                          |             |
> | &nbsp;&nbsp;Binding residues (mean ± SD)     | 1.412 ± 0.522            | --          |
> | &nbsp;&nbsp;Non-binding residues (mean ± SD) | 0.747 ± 0.347            | --          |
> | &nbsp;&nbsp;Ratio (binding/non-binding)      | 1.89×                    | --          |
>
> **We have added more detailed discussion in section 3.3, 4.4 and Appendix E of revised manuscript in blue color.**
>
> [1]. Liang, Jie, Clare Woodward, and Herbert Edelsbrunner. "Anatomy of protein pockets and cavities: measurement of binding site geometry and implications for ligand design." Protein science 7.9 (1998): 1884-1897.

---

> > ### Author Response · Authors · 2025-11-20
> > **Point-by-point responses to Reviewer TS28 (2/n)**
> >
> > > **W2.2 Expressive Power of Baselines**
> >
> > We agree with the reviewer that this is an important theoretical consideration. We have conduct the experiments with multiple layers of our method, and selected the best as mentioned in **Table-5 of Appendix F**. Regarding the baselines methods going sufficiently deep, current state of the method EquiPocket by going deeper (which we have compared with), establishes that **doing excessive message passing leads to oversmoothing, ultimately resulting into performance degradation**.  For GDEGAN, performance peaks at 4 layers (Average DCC=0.6050) and degrades with deeper architectures (5 layers: 0.588, 6 layers: 0.583), despite each additional layer adding nearly 20\% more parameters. This occurs because standard attention is fundamentally pairwise, computing similarity between node pairs independently, while variance requires neighborhood-level aggregation. Our Gaussian attention explicitly computes $\sigma_i^2$ at each layer, providing direct inductive bias. **Furthermore, our variance analysis (Appendix C.5) shows binding sites have 1.89× higher variance ($r=0.39$, $p<0.001$, $d=1.85$)**, confirming explicit computation captures biologically meaningful signals that standard attention cannot learn implicitly.
> >
> > **We have expanded Section 4.5 and added detailed results in Appendix G.**
> >
> > > **W3.1-Q3: Experimental Evaluation Issue (Training Stability)**
> >
> > We have conducted a comprehensive stability analysis comparing GDEGAN and GotelNet(full) as asked by the reviewer across training epochs until training converges from early-stopping criteria. The empirical deviance from the training and validation loss shows that $(\sigma_i)^2$ term in our GDEGAN not only maintains the stability but also improves training convergence compared to GotenNet(full). GDEGAN demonstrates **5.3\% lower training loss variance (0.0179 vs 0.0189)** compared to GotenNet(full), indicating more stable gradient flow despite $(\sigma_i)^2$ computation. Throughout training we observed monotonic convergence. Interestingly, while GDEGAN shows marginally higher validation loss variance, it achieves better test performance as indicated in **ablation Table-2** of the manuscript. This validation-test gap indicates that our method learns the right features for binding site localization rather than overfitting to loss minimization. Here we present the quantitative metrics for better reflection of training stability.
> >
> > | **Metric**                     | **GDEGAN** | **GotenNet (full)** |
> > |--------------------------------|------------|----------------------|
> > | Training loss variance         | 0.0179     | 0.0189               |
> > | Validation loss variance       | 0.00012    | 0.00010              |
> >
> > **We have updated the section 3.3 and updated Appendix-D with detailed analysis and plots in the revised manuscript.**

---

> > > ### Author Response · Authors · 2025-11-20
> > > **Point-by-point responses to Reviewer TS28 (3/n)**
> > >
> > > > **W3.2 and Q4: Parameter Sensitivity Analysis (Computational Overhead)**
> > >
> > > In the **Remark-2** of the manuscript we provide the theoretical computation complexity of our method and to be fare in comparison with all the existing state of the art methods, we have also reported the **inference time (in seconds) of our method per 100 proteins of varying length in Appendix(C.4) of the manuscript**. Our method takes around 1.9 seconds, in contrast to GotenNet’s 4.12 seconds and Equipocket’s 37 seconds per 100 proteins. This shows the computation efficiency of **our method making it approx 50\% faster than GotenNet and approx 95\% faster than EquiPocket.**
> > >
> > > As suggested, we have also performed the experiment to measure the wall-clock training time overhead for the Gaussian Dynamic Attention layer and dot-product attention layer in the GotenNet. We report the average attention mechanism computation time (in milliseconds) overhead for proteins of varying sizes.
> > >
> > >
> > > | **Protein Size**            | **Gaussian Attention (ms)** | **Dot-product (ms)** |
> > > |-----------------------------|------------------------------|------------------------|
> > > | Nodes ≤ 100                 | 0.26                         | 0.57                   |
> > > | 100 < Nodes ≤ 500           | 0.28                         | 0.62                   |
> > > | Nodes > 500                 | 0.31                         | 0.68                   |
> > >
> > > We also analyzed learned temperature parameters across 5 independent training runs until the model convergence. Across all runs, heads specialized from near-uniform initialization (epoch 1)(std=0.059) to diverse final states (std=0.330), a 5.6 $\times$ increase in diversity. This reproducible pattern demonstrates that specialization is learned, not random. Temperatures smoothly decreased from mean 2.03 (epoch 1) to 1.82 (epoch 50), with stable convergence. This indicates the network learned that more selective attention $\xi$ is beneficial for localized binding sites. Final leaned temperatures from the best models across all runs spans from 1.456 to 2.318, with different values for each head.
> > > **We have added the discussion in Section 4.5 (line 485-512) and have also added the visualization in Figure 4b and 4c of the revised manuscript.**

---

> ### Author Response · Authors · 2025-11-24
> **Looking forward to your feedback.**
>
> Dear Reviewer TS28,
>
> Thank you once again for your valuable feedback and thoughtful comments.
>
> In our rebuttal, we have provided additional experiments, analyses, and clarifications, and we sincerely hope that they address the points you raised.
>
> If there are any remaining questions or concerns, we would be more than happy to provide further clarification.
>
> Should our responses adequately address the issues, we would be grateful if you could kindly reconsider the evaluation and the scores.
>
> Thank you for your time and thoughtful evaluation.
>
> The Authors

---

> > ### Author Response · Authors · 2025-11-28
> > **Eagerly waiting for post-rebuttal feedback**
> >
> > Dear Reviewer TS28,
> >
> > Thank you once again for your insightful and thoughtful comments!
> >
> > We were encouraged by your recognition of our work's novelty, particularly your observation that our approach provides "a more physically grounded inductive bias" captures precisely the motivation behind our methods design.
> >
> > We appreciate the critical examination regarding the mathematical equations, hypothesis justification, and training stability; addressing these points pushed us to provide more comprehensive analyses that we believe significantly improve the manuscript.
> >
> > As the reviewer-author discussion period is closing soon, we wanted to gently follow up to see if you have had the opportunity to review our response, and whether any aspects require further clarification.
> >
> > We have updated our revised manuscript, where we made the following updates as per your suggestions (summary):
> >
> > - Mathematical clarification: We have **revised manuscript section 3.3 equation (5-8) to ensure reproducibility.**
> > - Justification of Core Hypothesis: **We have added detailed discussion in section 3.3, 4.4 and Appendix E of revised manuscript.**
> > - Training stability: **We have updated the section 3.3 and updated Appendix-D with detailed analysis and plots in the revised manuscript for training stability as you suggested.**
> > - Expressive Power of Baselines: **We have expanded Section 4.5 and added detailed results in new Appendix G** etc.
> >
> > We genuinely value your expertise in evaluating our work. If there are any remaining concerns or additional experiments that would strengthen your confidence further in our contribution, we would be very happy to address them. Should our responses adequately address the issues, we would be grateful if you could kindly reconsider the evaluation and the scores.
> >
> > We look forward to your response.
> >
> > Thank you,
> >
> > The Authors

---

### Official Review · Reviewer_4P1F · 2025-10-25

**Soundness:** 3
**Presentation:** 3
**Contribution:** 2
**Rating:** 4
**Confidence:** 3

**Summary:**

The authors propose a variation of vanilla dot product attention based on adaptive kernels as a motivation to capture local geometric and chemical features of the residues to predict protein-ligand binding sites. The approach calculates mean and variance of a residue's neighborhood on the fly. They show the approach combined with ESM embeddings as node features  significantly outperform previous SOTA GotenNet that was based on dot product attention.

The authors claim that current best ligand binding site prediction methods that are based on message passing networks use context-agnostic attention mechanisms and the binding sites in a protein are often clustered based on their local geometric and chemical features. They construct a protein structure graph with residues as nodes and edges between the residues determined by a threshold on C-alpha distances. And the third component of the graph is the C-alpha coordinates. They initialize the nodes using ESM-2 node embeddings and then project to hidden dimensional space using learned transformations. Nodes are labeled 1 or 0 based on closeness to ligand atoms. The task is given a protein graph, predict the binding probability of each residue. The authors adopt Gotennet and modify the attention part and representations for scalars and tensors. They design basis functions based on spherical harmonics to preserve equivariance. They then use these steerable features and the invariant scalars from the projected pLM embeddings to create the message passing networks for both nodes and edges. The node features go through a dot product with the RBF features ensuring differentiability. Steerable features are initialized to 0 and then updated during training. The node features (from pLM) are used to compute mean and variance on the fly.

Experiments are conducted on three benchmark datasets and an additional ablation study is performed.

**Strengths:**

The paper is well organized in terms of the limitations of the current binding site prediction models. The adoption of GotenNet and modifying the vanilla attention with the proposed approach to make the attention more dynamic and aware of an atom's local neighborhood is a an interesting approach. The key contribution is the idea of computing a neighboring atom’s features from Gaussian distribution defined by the target atom’s local neighborhood in the model. The results in Table 1 show that GDEGAN beat the baseline models on three benchmark datasets.

**Weaknesses:**

From the results in Table 1 it shows the proposed method beats GotenNet on all three datasets except on the failure rate. However, from the ablation study shows (in Table 2) that  the main boost comes from the ESM embeddings for both methods. As the authors show that the proposed approach is beneficial as structural heterogeneity increases. Since protein ligand binding site is determined by the chemical fingerprint it would be interesting to see if the method relied no only on the C-alpha atoms but an all atom graph model like GearBind.

If that is a significant stretch the authors could also try using embeddings from structure aware protein models such as Prostt5 or SaProt or even surface-structure aware protein models such as AtomSurf ? If the main hypothesis is that the Gaussian kernels give the additional boost in performance by capturing the local chemical and geometric features then one could test that by extracting residue embeddings from the advanced protein models that are trained on structure and surface features constructed from local neighborhoods. Without that comparison it is hard to determine if the proposed approach is the optimal method to capture local geometric and chemical characteristics of the binding residues.

In any case when referring to chemical and geometric features the authors did not cite a few other relevant papers:
1. MaSIF: https://www.biorxiv.org/content/10.1101/606202v1.full.pdf
2. AtomSurf: Which combines structure and surface (https://arxiv.org/pdf/2309.16519)
3. GearBind: https://www.biorxiv.org/content/10.1101/2023.08.10.552845v1

**Questions:**

Please add other SoTA methods on protein ligand binding tasks such as HoloProt (https://arxiv.org/pdf/2204.02337)

---

> ### Author Response · Authors · 2025-11-20
> **Point-by-point responses to Reviewer 4P1F (1/n)**
>
> Dear Reviewer 4P1F,
>
> We thank you for acknowledging the strengths of our research and the valuable feedback. We are happy that you found our approach interesting. We sincerely appreciate the detailed comments and questions. Below we Provide the point-by-point response to each of the comments and questions asked.
>
> > **W1: Failure rate, ESM Embeddings and all atom graph**
>
> 1. We sincerely apologise for the confusion caused by our presentation in Table-1. We made the presentation more clear to avoid any further confusion to the readers. We would like to clarify thet in Table-1 of the manuscript, GDEGAN actually achieves a **lower failure rate (3.2\% compared to GotenNet (4.9\%), representing a relative improvement**.
>
> 2. The point about chemical fingerprint is important and we would like to clarify our design choices and their implications. **ESM embeddings [1] already provides are learned compression of atomic level chemical information into residue level representation through evolutionary learning**. This makes ESM embeddings not only sufficient but optimal for our case. Furthermore, we have already compared with state-of-the-art methods such as EquiPocket, which uses full atom level graphs and our methods GDEGAN outperforms EquiPoeckt by 42.36\% in DCC and 13.73\% in DCA (averaged across datasets) and because our method uses residue level graph it also outperforms EquiPocket while inference by approx 95\% as shown in Appendix C.4. The residue level representation of proteins makes them approximately 10 times smaller graph size (hundreds of residues vs thousands of atoms), which makes our approach even more scalable.
>
> > **W2: Embeddings from structure aware methods**
>
> Following the suggestion to test with ProstT5 [2] or similar structure-aware embeddings addresses an important question, but there is a critical distinction: our Gaussian Dynamic Attention computes adaptive statistics $\mu_i$ and $\sigma_i^{2}$ at each layer based on evolving learned features, while structure-aware embeddings provide fixed geometric encoding at the input layer. These represent fundamentally different approaches to incorporating geometric information.  **Fixed Structural Encoding (ProstT5, SaProt):** These methods encodes local neighborhoods and geometric properties as static features derived from 3D coordinates during pretraining. These representations remain constant throughout the network.  **Adaptive Statistical Attention (Ours):** Computes neighborhood statistics dynamically at each message-passing layer from learned features. The variance $\sigma_i^{2}$ that modulates attention in layer $l+1$ is computed from features $l$, allowing it to capture emergent chemical and geometric patterns that develop through learning.
>
> Following your recommendation, we evaluated GDEGAN with ProstT5 embeddings [2], which already encode structural features along with surface accessibility information from pretraining on protein structures.
>
> | **Methods**          | **COACH420 (DCC ↑)** | **COACH420 (DCA ↑)** | **HOLO4K (DCC ↑)** | **HOLO4K (DCA ↑)** | **PDBbind2020 (DCC ↑)** | **PDBbind2020 (DCA ↑)** |
> |----------------------|---------------------|---------------------|-------------------|-------------------|-------------------------|-------------------------|
> | **GDEGAN + ProstT5** | 0.521 (0.002)       | 0.655 (0.001)       | 0.489 (0.009)     | 0.720 (0.006)     | 0.601 (0.009)           | 0.752 (0.010)           |
> | **GDEGAN + ESM**     | **0.580 (0.008)**   | **0.707 (0.009)**   | **0.560 (0.013)** | **0.788 (0.011)** | **0.675 (0.010)**       | **0.826 (0.011)**       |
>
> The results provide evidence that our approach is the optimal method to capture local geometric and chemical characteristics of the binding residues.
> As shown in the table, GDEGAN with ProstT5 performs 10.2\% worse than with ESM embeddings (DCC averaged across datasets). This result aligns with AtomSurf (ICLR 2025) [3], where they combined surface representations with graphs and found that even with sophisticated bipartite message passing, surface methods under-performs.
> **They also showed that structure-aware information in embeddings can hurt generalization in structure aware methods.** The performance drop with ProstT5 confirms the information bottleneck principle [4], which states that encoding structure in both embeddings and geometric processing creates harmful interference. ProstT5's fixed structural encoding conflicts with our method's need to learn adaptive geometric representations through message passing. The network cannot effectively learn to compute meaningful variance statistics when the input already encodes rigid geometric neighborhoods.
>
> Combined with our ablation study shown in Table 2 of the manuscript, where **Gaussian dynamic attention improves by 5.3\% compared to dot-product attention even with identical ESM embeddings**, these experiments establish that our architectural contribution is the primary driver of performance.

---

> > ### Author Response · Authors · 2025-11-20
> > **Point-by-point responses to Reviewer 4P1F (2/n)**
> >
> > > **W2: Embeddings from structure aware methods (continue)**
> >
> > **Why ESM-2 is the Optimal Choice for Our Method:**
> > ESM-2 provides orthogonal information to our geometric processing: sequence context and evolutionary patterns without encoding 3D structure. This allows our Gaussian Dynamic Attention to freely learn adaptive geometric representations through message passing without interference from conflicting fixed encoding. The fact that structure-aware embeddings (ProstT5 with 3B parameters) perform worse than sequence-only embeddings (ESM-2 with 650M parameters) further validates our design principle.
> >
> > **Adding relevant citations:**
> > We have now incorporated citations to AtomSurf and GearBind in the revised manuscript. We would like to note that MaSIF was already cited in our Introduction (line 50). Additionally, following your insightful feedback, **we have added a discussion on our choice of ESM-2 embeddings versus structure-aware alternatives in Section 3.2 (line 198-200)**.
> >
> > > **Q: Adding other SoTA methods like HoloProt on PLB tasks.**
> >
> > We really appreciate the reviewers suggestion to compare against HoloProt [5]. We also agree that this is the highly relevant and impactful method in protein-ligand tasks where, protein ligand both are known i.e. binding affinity prediction.
> >
> > However, we respectfully note that HoloProt [5] addresses protein-ligand binding affinity prediction, where the goal is to predict the binding strength between a known protein-ligand pair. This is a regression task that requires both the **protein structure AND the ligand molecule as input**, outputting a continuous affinity value. Whereas, our work GDEGAN addresses binding site identification, where the goal is to identify potential binding regions on a protein surface **without knowing the ligand**. This is a classification task that takes only the protein structure as input and outputs binary classifications for each residue (binding/non-binding).
> > **We have cited the HoloProt and have ensured our manuscript clearly distinguishes between these two important but distinct tasks to guide future readers (Introduction, line 52-53 in revised manuscript)**.
> >
> > [1]. Lin, Zeming, et al. ``Language models of protein sequences at the scale of evolution enable accurate structure prediction." BioRxiv 2022 (2022): 500902.
> >
> > [2]. Heinzinger, Michael, et al. "ProstT5: Bilingual language model for protein sequence and structure." NAR Genomics and Bioinformatics 6.4 (2024).
> >
> > [3]. Mallet, Vincent, et al. "Atomsurf: Surface representation for learning on protein structures." arXiv preprint arXiv:2309.16519 (2023).
> >
> > [4]. Tishby, Naftali, and Noga Zaslavsky. "Deep learning and the information bottleneck principle." 2015 ieee information theory workshop (itw). Ieee, 2015.
> >
> > [5]. Somnath, Vignesh Ram, Charlotte Bunne, and Andreas Krause. "Multi-scale representation learning on proteins." Advances in Neural Information Processing Systems 34 (2021): 25244-25255.

---

> ### Author Response · Authors · 2025-11-24
> **Looking forward to your feedback.**
>
> Dear Reviewer 4P1F,
>
> Thank you once again for your valuable feedback and thoughtful comments.
>
> In our rebuttal, we have provided additional experiments, analyses, and clarifications, and we sincerely hope that they address the points you raised.
>
> If there are any remaining questions or concerns, we would be more than happy to provide further clarification.
>
> Should our responses adequately address the issues, we would be grateful if you could kindly reconsider the evaluation and the scores.
>
> Thank you for your time and thoughtful evaluation.
>
> The Authors

---

> > ### Comment · Reviewer_4P1F · 2025-11-26
> >
> > Thank you for adding the two benchmarks and addressing that the proposed method does not require knowledge of the ligand. While it would be interesting to compare performance in the presence, the new baselines help me to raise my score to 6

---

> ### Author Response · Authors · 2025-11-26
> **Response to Reviewer 4P1F Feedback**
>
> Dear Reviewer 4P1F,
>
> We sincerely thank you for your positive feedback and for raising the score to 6 (changes does not reflect in our page yet).
>
> We agree that the comparison in presence of the ligand would be an interesting work, but in the current work we are addressing the finding out of the probable binding sites. We strongly believe that our method could be used for various other down streaming task such as affinity prediction and docking etc.
>
> If you feel that additional experiments or results within the scope of the current problem statement would further strengthen the manuscript, we are more than willing to provide them and actively engage in any further discussions.
>
> We would be grateful if you could review the revised version and let us know if it is possible to change the decision on borderline accept, along with your view on the soundness, presentation and contributions as well.
>
> Thank you once again for your valuable insights and support. We look forward to your response.
>
> The Authors

---

### Official Review · Reviewer_trg9 · 2025-10-27

**Soundness:** 3
**Presentation:** 2
**Contribution:** 2
**Rating:** 4
**Confidence:** 3

**Summary:**

The paper introduces GDEGAN, a Gaussian Dynamic Equivariant Graph Attention Network for protein–ligand binding site prediction. Its central idea is to replace dot-product similarity with a Gaussian kernel whose weights are determined by local neighborhood statistics and a learnable temperature, implemented within an SE3-equivariant graph architecture. The approach targets the strong geometric and chemical heterogeneity of protein surfaces and includes a clear treatment of symmetry, noting that the use of ESM-2 scalar features yields SE3 rather than full E3 equivariance. The training objective addresses class imbalance and directional cues. Experiments on COACH420, HOLO4K, and PDBbind2020 report consistent gains in DCC and DCA, substantial reductions in failure rate, and faster inference versus strong baselines. Attention visualizations align with pocket regions, offering an interpretable account of model behavior.

**Strengths:**

Strength 1：Uses a local Gaussian kernel with adaptive bandwidth from neighborhood statistics and a learnable temperature, yielding context-aware attention suited to heterogeneous protein surfaces.
Strength 2：Provides formal analysis showing the proposed attention preserves SE(3) equivariance under the chosen feature representation, giving a clear geometric justification.
Strength 3 ：Demonstrates consistent improvements over strong baselines across standard pocket benchmarks, supported by ablations and qualitative visualizations, with competitive or better inference efficiency.

**Weaknesses:**

Weakness 1： The paper claims to capture geometric structure and handle variation among neighboring residues, but the evidence is mostly indirect. It should demonstrate which previously hard geometric challenges are now addressed, with targeted analyses rather than only aggregate metrics and visuals.
Weakness 2：Comparisons with prior graph-attention variants are incomplete, especially kernelized attention methods. A deeper analysis against these baselines is needed to substantiate the claimed contribution and clarify what is genuinely new.
Weakness 3：Key terms should be standardized for readability, including “Gaussian kernel,” “Gaussian attention,” and “Protein-aware Structural Embeddings.” A thorough pass is recommended. Also correct the misspelling of “temperature” in the figures.
Weakness 4：The proposed variant may introduce extra computational cost. The paper should provide complexity measurements and hyperparameter sensitivity analyses to quantify overhead and practical trade-offs.

**Questions:**

Q1：Can you report how the learnable temperature evolves and distributes during training?
Q2: How does using a learnable temperature compare to a fixed bandwidth parameter?
Q3：Which specific geometric structures previously handled poorly by dot-product or standard graph attention are now captured better?
Additional questions and suggestions please refer to the Weaknesses.

---

> ### Author Response · Authors · 2025-11-20
> **Point-by-point responses to Reviewer trg9 (1\n)**
>
> Dear Reviewer trg9,
>
> We thank you for acknowledging the strengths of our research and the valuable feedback.  We sincerely appreciate the detailed comments and questions. Below we Provide the point-by-point response to each of the comments and questions asked.
>
> > **W1-Q3: Which previously hard geometric challenges are now addressed ?**
>
> We acknowledge that our initial presentation relied more on aggregate metrics and qualitative visualizations. We have now added targeted quantitative analyses following reviewer suggestion that directly demonstrate which specific geometric challenges our method addresses.
>
> Binding pockets are characterized by distinct geometric signatures including concave curvature, depth variations, and spatial clustering of diverse residue types, which create local heterogeneity—neighbouring residues within binding sites exhibit greater diversity in spatial arrangements and chemical environments compared to uniform surface patches [1].
>
> 1. The fundamental geometric challenge we address is identifying sharp transitions between binding and non-binding regions, which are characterized by abrupt changes in local chemical and geometric properties. Standard dot-product attention treats all neighborhoods uniformly, using a globally fixed similarity metric that cannot adapt to local heterogeneity. Our learned neighbourhood variance $\sigma_i^2$ directly captures this **geometric heterogeneity**. We provide rigorous statistical validation in **Appendix E ** on 205,791 residues from the validation set. This analysis directly evaluates whether the learned neighbourhood variance $(\sigma_i^2)$ in our Gaussian Dynamic Attention mechanism correlates with ground-truth binding site locations. We observe a strong, statistically significant positive correlation (Pearson r = 0.3865, p $<$ 0.001) with a very large effect size (Cohen's d = 1.8545). Binding site residues exhibit 1.89 times higher learned variance compared to non-binding residues (mean: 1.412 vs 0.747, p $<$ 0.001). This demonstrates that local variance provides strong discriminative power for identifying binding site boundaries—**a property that uniform attention mechanisms cannot exploit**. The variance normalization directly amplifies attention at these high-variance boundaries (also shown in the attention plots Figure 3 of the manuscript), enabling precise boundary discrimination that was previously unattainable.
>
> 2. Our targeted analysis in **Section 4.5** on HOLO4K (Table 2) reveals that GDEGAN(full) achieves a 2.4\% DCC improvement over GotenNet(full), compared to only 1.5\% improvement on the more homogeneous COACH420 dataset. This differential performance directly demonstrates that statistical adaptation provides greater benefits as structural heterogeneity increases.
>
> **We have also added the discussion in section 3.3 (line 237-240) and 4.4 (line 443-451) in the updated manuscript.**
>
> [1]. Liang, Jie, Clare Woodward, and Herbert Edelsbrunner. "Anatomy of protein pockets and cavities: measurement of binding site geometry and implications for ligand design." Protein science 7.9 (1998): 1884-1897.

---

> > ### Author Response · Authors · 2025-11-20
> > **Point-by-point responses to Reviewer trg9 (2\n)**
> >
> > > **W2: Comparison Completeness.**
> >
> > Graph attention mechanism can be mainly categorized into three main paradigms:
> > **1. Dot-product attention:**  Within the graph neural network literature, methods based on standard graph attention [1], including Equivariant graph attention methods like EquiPocket [2] and GotenNet [3], employ dot-product attention that treats all neighborhoods uniformly. **2. Distance weighted attention:** Method like ScheNet [4], in which the continuous filter $\Tilde{W}(r_{ij})$ acts as an attention-like weight determined by a function (where that function uses a Gaussian basis for the distance, which is a common component in RBF/Gaussian kernels). **3. Kernelized attention:** Such methods in general use kernel functions (e.g. polynomial) for similarity computation, but break equivariance. Specifically, we did not find any other kernal based attention methods which does not break equivariance constrain and are applied to geometric feature data such as our case of protein-ligand binding site identification. We compare against representative from all categories including methods like GAT [1], Equipocket [2], GotenNet [3] and ScheNet [4] as shown in **Table 1 of our manuscript**.
> >
> > Our contribution is the adaptive statistical attention mechanism designed for equivariant geometric graphs, with protein-specific inductive biases (variance-based modulation, empirically shown in **Appendix E** ) and integration with high-degree steerable features. This represents a distinct advance over both general kernelized attention (which lacks equivariance and geometric specialization) and existing equivariant graph attention (which uses uniform similarity measures).
> >
> > [1]. Veličković, Petar, et al. "Graph attention networks." arXiv preprint arXiv:1710.10903 (2017).
> >
> > [2]. Zhang, Yang, et al. "Equipocket: an e (3)-equivariant geometric graph neural network for ligand binding site prediction." ICML (2023).
> >
> > [3]. Aykent, Sarp, and Tian Xia. "Gotennet: Rethinking efficient 3d equivariant graph neural networks." ICLR, 2025.
> >
> > [4]. Schütt, Kristof T., et al. "Schnet–a deep learning architecture for molecules and materials." The Journal of chemical physics 148.24 (2018).
> >
> > > **W3: key terms standardization and spelling correction in figures.**
> >
> > As suggested we have standardized the key terms in the revised manuscript. A complete thorough pass has been done to rectify any other mistakes including misspelling in the figures. All the changes are marked in the blue color in the revised manuscript.
> >
> > > **W4: Computational Cost and hyperparameter sensitivity analyses to quantify overhead.**
> >
> > We appreciate the opportunity to clarify and expand upon our computational analysis, which **was presented in Remark 2 (lines 341-347)** but may have been too concise. Our Gaussian Dynamic Attention achieves $O(d)$ fold complexity reduction (Remark 2), which is also empirically validated in the **Inference time comparison Appendix D (Table 3)** , showing GDEGAN processes 100 proteins in 1.90 seconds versus GotenNet's (dot-product) 4.12 seconds **(2.17× faster)**, confirming that our variance computation introduces minimal overhead while actually reducing total cost through simpler attention mechanism.
> >
> > Following the reviewer's suggestion, we have added comprehensive sensitivity analyses in **new Appendix G (Table 6) and Appendix D (Figure 5)**. Our layer depth analysis **Section 4.5 (figure 4)** reveals performance peaks at 4 layers (average DCC = 0.605) and degrades with additional depth (6 layers: DCC = 0.583), demonstrating robust performance within 3-5 layers with predictable trade-offs beyond this range. Additionally, our training stability analysis shows that variance-based attention actually improves optimization showing **5.3\% lower training loss variance (0.0179 vs 0.0189 for GotenNet)** and monotonic convergence.
> >
> > > **Q1: How learnable temperature evolves and distributes during training ?**
> >
> > We analyzed learned temperature parameters across 5 independent training runs until the model convergence. Across all runs, heads specialized from near-uniform initialization (epoch 1) (std=0.059) to diverse final states (std=0.330), a 5.6 $\times$ increase in diversity. This reproducible pattern demonstrates that specialization is learned, not random. **Temperatures smoothly decreased from mean 2.03 (epoch 1) to 1.82 (epoch 50), with stable convergence**. This indicates the network learned that more selective attention $\xi$ is beneficial for localized binding sites. Final leaned temperatures from the best models across all runs spans from **1.456 to 2.318**, with different values for each head.
> >
> > **We have added the discussion in Section 4.5 (line 485-512) and have also added the visualization in Figure 4b and 4c in the updated manuscript.**

---

> ### Author Response · Authors · 2025-11-20
> **Point-by-point responses to Reviewer trg9 (3\n)**
>
> > **Q2: Learnable vs fixed bandwidth comparison.**
>
> We trained GDEGAN to see how does learnable temperature compares with the fixed temperature, under identical architecture, training procedure, and hyper-parameters, with only the temperature parameter fixed vs. learnable.
>
> We have initialized all the heads temperature with the value 2.0, in which we have got the best results reported in the manuscript Table 1 (learned temperature).
> While **fixing the temperature value**, the average DCC across all the datasets **decreases about 6\% (DCC from 0.605 to 0.570)** demonstrating that a single fixed temperature across all the heads can not handle multi-scale binding sites heterogeneity inherited in protein binding sites. This performance gap arises because fixed temperature forces all 8 attention heads (same value as 2.0 for all) to use identical Gaussian kernel widths, resulting in redundant attention patterns, whereas learnable temperature enables head specialization—as evidenced by our analysis showing **learned values of temperature spanning [1.456, 2.318] with 5.6 $\times$ increased diversity (std: 0.059 to 0.330).** Different heads learn to focus on different geometric scales. This complementary multi-scale specialization, achieved automatically through gradient-based optimization without manual tuning, validates learnable temperature as a fundamental design choice.

---

> ### Author Response · Authors · 2025-11-24
> **Looking forward to you feedback.**
>
> Dear Reviewer trg9,
>
> Thank you once again for your valuable feedback and thoughtful comments.
>
> In our rebuttal, we have provided additional experiments, analyses, and clarifications, and we sincerely hope that they address the points you raised.
>
> If there are any remaining questions or concerns, we would be more than happy to provide further clarification.
>
> Should our responses adequately address the issues, we would be grateful if you could kindly reconsider the evaluation and the scores.
>
> Thank you for your time and thoughtful evaluation.
>
> The Authors

---

> > ### Comment · Reviewer_trg9 · 2025-11-25
> > **Response to Authors**
> >
> > The author's response resolved some of my important questions, and I will raise my score to 6.

---

> ### Author Response · Authors · 2025-11-25
> **Response to Reviewer trg9 Feedback**
>
> Dear Reviewer trg9,
>
> We sincerely thank you for your positive feedback and for raising the score to 6. We have carefully incorporated all your suggested changes into our revised manuscript. We have also done the thorough pass along with the highlighted corrections raised by you, which helped us in making the manuscript presentation better.
>
> If you feel that additional experiments or results would further strengthen the manuscript, we are more than willing to provide them and actively engage in any further discussions.
>
> We would be grateful if you could review the revised version and let us know if it is possible to change the decision on borderline accept, along with your view on the soundness, presentation and contributions as well.
>
> Thank you once again for your valuable insights and support. We look forward to your response.
>
> The Authors

---

### Author Response · Authors · 2025-11-29
**Summary of Responses and Revised Manuscript by Authors**

Dear AC / SACs,

We sincerely thank the reviewers for taking the time to provide thoughtful feedback, comments, and questions. We have revised the main manuscript and appendix, with all changes highlighted in blue. We would like to highlight the positive feedback we received from the reviewers:

---

- GDEGAN was recognized as a **"distinct and impactful innovation"** providing **"a more physically grounded inductive bias"** for protein-ligand binding site identification.
- Reviewers appreciated the **novelty** of adapting a **"probabilistic, variance-aware attention mechanism"** to 3D equivariant graph representations.
- The **experimental results** were acknowledged as **"substantial and consistent improvements"** with **"compelling"** relative gains (37-66% DCC).
- Reviewers recognized our **formal theoretical analysis** showing SE(3) equivariance preservation, providing **"clear geometric justification"**.
- The **context-aware attention design** using adaptive bandwidth from neighborhood statistics was found **"suited to heterogeneous protein surfaces"**.

Additionally, we believe we have thoroughly addressed the reviewers’ concerns offering new results and clarifications. Below we provide a concise summary of key concerns raised and how we addressed them:

---

### **Key Updates**

-  **Mathematical Ambiguity Clarification**: We revised **Section 3.3 (Equations 5-8)** providing more clarification to ensure reproducibility of our Gaussian Dynamic Attention mechanism, by how element-wise variance is computed and used for dimension-wise normalization followed by L2 norm.

-  **Empirical Validation of Core Hypothesis**: We conducted comprehensive statistical analysis on 205,791 residues showing binding sites exhibit 1.89× higher learned variance (Pearson r=0.39, Cohen's d=1.85, p<0.001) as suggested by reviewers. This validates our hypothesis that local feature variance is a meaningful signal for binding site identification. Added in **Section 3.3, 4.4 and Appendix E**.

-  **Training Stability Analysis**: We provided additional training and validation loss curves demonstrating GDEGAN achieves 5.3% lower training loss variance than GotenNet, confirming stable convergence despite variance computation. Added in **Section 3.3 and Appendix D**.

-  **Computational Overhead**: We provided wall-clock measurements showing Gaussian attention (0.26-0.31ms) is faster than dot-product attention (0.57-0.68ms), added in **Appendix C.4**. Overall, GDEGAN is 50% faster than GotenNet and 95% faster than EquiPocket during inference (**Appendix-D, Table-3**).

-  **Temperature Parameter Analysis**: We analyzed learned temperature evolution across training, showing heads specialize from uniform initialization to diverse final states. Added visualization in **Figure 4b and 4c**.

-  **Layer Depth Analysis**: We demonstrated performance peaks at 4 layers and degrades with deeper architectures due to oversmoothing, showing standard attention cannot implicitly learn variance-based patterns. Added in **Section 4.5 and Appendix G**.

-  **Additional Embedding Experiments**: Following reviewer suggestion, we evaluated GDEGAN with ProstT5 embeddings. Results show ESM-2 performs 10.2% better, validating our design choice of using sequence embeddings with adaptive geometric attention. Added in **Section 3.2**.

-  **Problem Scope Clarification**: We clarified that our task (binding site identification with protein-only input) is fundamentally different from binding affinity prediction (requires protein+ligand input), explaining why certain suggested comparisons are outside our scope. Added in **Introduction (lines 52-53)**.

-  **Presentation Improvements**: We standardized key terminology, fixed spelling typos in figures, and improved overall manuscript clarity. We have also added the additional citations asked by the reviewers.

---

### **Summary of Contributions**

- Novel Gaussian Dynamic Attention preserving SE(3) equivariance.
- 37-66% DCC improvement, 7-19% DCA improvement over state-of-the-art.
- 95% faster inference than EquiPocket.
- Comprehensive ablation studies validating each component with theoretical and empirical justification.

---

We kindly request you to consider our submission in light of the feedback received and the improvements made to our manuscript. We believe that we have addressed all major concerns raised by the reviewers, which has significantly improved the clarity and reliability of our manuscript.

Thank you for your time, consideration and your valuable service to the research community, especially given the unusual circumstances of this review cycle.

Sincerely,

The Authors

---

### Note · Authors · 2026-06-09

I have read and agree with the venue's withdrawal policy on behalf of myself and my co-authors.

---

### Meta-Review · Area_Chair_jp9h · 2026-01-06

**Summary:**

This paper proposes GDEGAN, a Gaussian Dynamic Equivariant Graph Attention Network for protein–ligand binding site prediction, motivated by the limitations of standard dot-product attention in capturing local chemical and geometric variation. By using adaptive, statistics-based Gaussian attention, the method reports improved performance on COACH420, HOLO4k, and PDBBind2020, with potential relevance for accelerating structure-based drug discovery.

Strengths:

The paper introduces Gaussian Dynamic Equivariant Graph Attention as a modification of standard dot-product attention for ligand binding site prediction. From the rebuttal, the requested hyperparameter ablation studies were addressed in a reasonably adequate manner, indicating that the authors engaged with at least part of the reviewers’ empirical concerns.

Weaknesses:

The main limitation is that the architecture closely follows GotenNet, and the connection between Gaussian attention and GotenNet’s distinctive components, such as spherical scalarization, is not fully clarified, making the proposed modification appear somewhat generic. The empirical evaluation would also benefit from including additional strong baselines, such as VN-EGNN on COACH420, to provide a more complete comparison. Finally, some parts of the presentation closely parallel GotenNet and could be revised to better highlight the paper’s original contributions.

**Reviewer Concerns:**

While some reviewers felt the manuscript addressed certain questions, the core issues remained unresolved. Furthermore, there were numerous crucial points (see weakness) that the reviewers, though not explicitly pointed out, that the authors should improve to enhance the manuscript's quality for resubmission.

**Reviewer Scores:**

Although some reviewers raised their scores (expressing their willingness), it appears that the core contributions of this paper still need to be supplemented and it has not yet reached a publishable level.

---

### Decision · Program_Chairs · 2026-01-26

Reject